# Nonlinearity in the Tropospheric Pathway of ENSO to the North Atlantic

Bernat Jiménez-Esteve[1] and Daniela I.V. Domeisen[1]

[1]Institute for Atmospheric and Climate Science, ETH Zurich, Universitätstrasse 16, 8092 Zurich, Switzerland

**Correspondence:** Bernat Jiménez-Esteve (bernat.jimenez@env.ethz.ch)

**Abstract.** El Niño Southern Oscillation (ENSO) can exert a remote impact on North Atlantic and European (NAE) winter climate. This teleconnection is driven by the superposition and interaction of different influences, which are generally grouped into two main pathways, namely the tropospheric and stratospheric pathways. In this study, we focus on the tropospheric pathway through the North Pacific and across the North American continent. Due to the possible non-stationary behaviour and the limited time period covered by reanalysis data sets, the potential nonlinearity of this pathway remains unclear. In order to address this question, we use a simplified physics atmospheric model forced with seasonally varying prescribed sea surface temperatures (SST) following the evolution of different ENSO phases with linearly varying strength at a fixed location. To isolate the tropospheric pathway the zonal mean stratospheric winds are nudged towards the model climatology. The model experiments indicate that the tropospheric pathway of ENSO to the North Atlantic exhibits significant nonlinearity with respect to the tropical SST forcing, both in the location and amplitude of the impacts. For example, strong El Niño leads to a significantly stronger impact over the North Atlantic Oscillation (NAO) than a La Niña forcing of the same amplitude. For La Niña forcings, there is a saturation in the response, with no further increase in the NAO impact even when doubling the SST forcing, while this is not the case for El Niño. These findings may have important consequences for long-range prediction of the North Atlantic and Europe.

## 1 Introduction

El Niño Southern Oscillation (ENSO) is the most important mode of interannual variability in the tropical Pacific. Trade winds, sea level pressure (SLP), precipitation, and sea surface temperatures (SST)s irregularly oscillate between a warm (El Niño, EN) and cold (La Niña, LN) phase (e.g., Philander, 1990; Diaz et al., 2001). By means of Rossby wave trains, the associated tropical circulation anomalies can also influence the extratropical circulation (Hoskins and Karoly, 1981; Sardeshmukh and Hoskins, 1988; Liu and Alexander, 2007). An important ENSO teleconnection is observed during winter over the North Pacific and the North American continent (Bjerknes, 1969; Horel and Wallace, 1981; Mo and Livezey, 1986; Ropelewski and Halpert, 1987; Halpert and Ropelewski, 1992; Trenberth et al., 1998), consisting of a strengthened Aleutian low (AL) (e.g. Barnston and Livezey, 1987) and a southward shift and eastward extension of the tropospheric jet and the storm track (Seager et al., 2010) during EN, projecting on the positive phase of the Pacific North American (PNA) pattern (Wallace and Gutzler, 1981).

Opposite-signed anomalies tend to be observed during LN, yet there are significant nonlinearities in the location and strength of the impacts (e.g., Zhang et al., 2014; Frauen et al., 2014; Garfinkel et al., 2018; Jiménez-Esteve and Domeisen, 2019).

The North Pacific circulation anomalies associated with ENSO can influence the North Atlantic. The ENSO signal reaches the North Atlantic through different mechanisms (e.g., Brönnimann, 2007; Li and Lau, 2012a; Jiménez-Esteve and Domeisen, 2018, and references therein), generally leading to the negative (positive) phase of the North Atlantic Oscillation (NAO) during

EN (LN) years and thus impacting winter temperature and precipitation over Europe (e.g, Ineson and Scaife, 2009). These mechanisms are generally grouped into two main pathways, namely the tropospheric and stratospheric pathways (e.g., Bell et al., 2009; Cagnazzo and Manzini, 2009; Ineson and Scaife, 2009; Butler and Polvani, 2011; Butler et al., 2014). Due to a less direct impact and the large internal variability over the North Atlantic, the ENSO teleconnection to the North Atlantic is less robust, and less stationary (Greatbatch, 2004; Deser et al., 2017, 2018; Garfinkel et al., 2019) than that to the North Pacific,

with a distinct response in early and late winter (Moron and Gouirand, 2003; Ayarzagüena et al., 2018). In the present study, the tropospheric pathway of the ENSO teleconnection to the North Atlantic is revisited using targeted atmospheric model experiments in order to obtain more robust statistics of this teleconnection and to analyze potential nonlinearities.

ENSO impacts the winter Arctic stratosphere, with a generally weaker (stronger) polar vortex during EN (LN) (e.g., Sassi et al., 2004; García-Herrera et al., 2006; Manzini et al., 2006; Manzini, 2009; Bell et al., 2009; Cagnazzo and Manzini, 2009;

Garfinkel and Hartmann, 2008; Iza et al., 2016; Domeisen et al., 2019). While observations and models generally agree on the sign of the winter mean stratospheric response, changes in the frequency of sudden stratospheric warmings (SSWs) are less robust (e.g., Butler and Polvani, 2011; Garfinkel et al., 2012; Polvani et al., 2017; Domeisen et al., 2019). For a summary of the results obtained in these studies see Table 2 in Trascasa-Castro et al. (2019). These results are sensitive to the dataset, the time period, and the classification of the SSW events, especially for LN, and hence time series longer than the current period

available for reanalysis data are necessary in order to make robust conclusions about the ENSO-SSW relationship (Weinberger et al., 2019). On average, Arctic stratospheric anomalies can exert a downward impact on the troposphere (Kidston et al., 2015, and references therein), especially over the Arctic and in the North Atlantic region (Baldwin and Dunkerton, 2001). A weaker (stronger) stratospheric polar vortex tends to be associated with a negative (positive) phase of the North Atlantic Oscillation (NAO), which impacts temperature and precipitation anomalies in the North Atlantic region (Baldwin and Dunkerton, 2001;

Ineson and Scaife, 2009; Cagnazzo and Manzini, 2009). Recently, using different atmospheric models Rao and Ren (2016a, b) and Trascasa-Castro et al. (2019) have identified nonlinearities in the stratospheric and North Atlantic response to ENSO, while (Weinberger et al., 2019) find a linear stratospheric response.

The tropospheric pathway of the ENSO teleconnection to the North Atlantic involves different mechanisms. In this study, we focus on the North Pacific downstream effect. An additional tropospheric pathway through the tropical Atlantic has been

proposed (e.g., Sung et al., 2013; Rodríguez-Fonseca et al., 2016), which might be particularly relevant for strong EN events (Toniazzo and Scaife, 2006; Bell et al., 2009; Hardiman et al., 2019). In the troposphere, quasi-stationary (QS) waves and transient eddies propagate eastward from the North Pacific to the North Atlantic, influencing the circulation in the North Atlantic sector (e.g., Weare, 2010; Li and Lau, 2012a, b; Drouard et al., 2013, 2015; Jiménez-Esteve and Domeisen, 2018; Schemm et al., 2018). This downstream effect from the North Pacific can be divided into the following mechanisms: First, a

direct increase (decrease) in the downstream propagation of transient eddies across North America during EN (LN) conditions generally leads to negative (positive) SLP anomalies in the southern lobe of the NAO (Li and Lau, 2012a, b) that is associated with a southward (northward) shift of the North Atlantic tropospheric eddy-driven jet. A second mechanism acts through remote baroclinicity changes over North America and the west North Atlantic associated with the PNA phase, which can influence the North Atlantic circulation (Pinto et al., 2011). Such changes in baroclinicity and the background flow affect the genesis of extratropical cyclones over the Rocky mountains, Greenland, and the Gulf stream region (Schemm et al., 2018). Third, Drouard et al. (2013, 2015) highlight the role of the meridional tilt of the transient eddies propagating along the jet stream, affecting the type of wave breaking in the North Atlantic during different ENSO phases. Finally, large-scale QS waves also flux energy eastward, which is important for linking the low frequency AL variability with the Icelandic low variability (Honda and Nakamura, 2001; Honda et al., 2005; Orsolini et al., 2008), although this link shows strong decadal variability (e.g., Honda and Nakamura, 2001; Sun and Tan, 2013). Jiménez-Esteve and Domeisen (2018) show that the eastward propagation of large-scale (zonal wavenumbers 1-3) QS waves generally increases (decreases) during LN (EN) and/or during strong (weak) vortex events.

In summary, the described mechanisms constituting the tropospheric pathway of ENSO through the North Pacific are: changes in the total eastward wave activity fluxes of transient and QS waves, remote changes of baroclinicity and cyclogensis, and changes in the frequency of the type of wave breaking in the North Atlantic. These mechanisms are not independent and are related for example through the PNA phase. Therefore, the tropospheric pathway for the ENSO impact on the North Atlantic is driven by a combination of the previous effects modulated through the North Pacific variability. Here, we focus on the effect of the changes in the total eastward wave activity fluxes (WAF) of transient and QS waves, and the baroclinicity mechanism. We find that nonlinearities in the North Atlantic response to ENSO are better explained in terms of the WAF mechanism.

The stratospheric and tropospheric pathways are also not independent and therefore their respective impacts in the North Atlantic cannot be clearly separated using reanalysis datasets due to the small sample size when subdividing ENSO events into different polar stratosphere states (e.g., Polvani et al., 2017; Jiménez-Esteve and Domeisen, 2018). Garfinkel et al. (2012) find that ENSO only accounts for some 10% of vortex variability, and that the correlation between Nino3.4 and seasonal mean vortex strength is ∼0.3 in reanalysis as well as in a range of models. This is supported by Polvani et al. (2017), who report that approximately 90% of all SSWs are not directly caused by ENSO, and anomalous polar vortex states can occur in either phase of ENSO due to the large internal variability of the polar stratosphere. Also, despite both EN and SSW projecting onto a negative NAO, the surface impacts are not exactly the same (Oehrlein et al., 2019) and they should be considered independently. To resolve the question of the relative importance of the stratospheric and tropospheric pathways, Bell et al. (2009) compared model simulations with a degraded and not degraded stratosphere and found a distinct and less zonal surface response for a strong EN forcing when the stratospheric pathway is strongly suppressed. In agreement with this finding, Cagnazzo and Manzini (2009) found that low-top models cannot fully reproduce the stratospheric pathway due to their inability to reproduce the SSW increase during EN and therefore the full response in the NAE sector. Here we extend the results of the previous studies, and use idealized atmospheric model experiments forced with ENSO-like SST forcing as in Jiménez-Esteve and Domeisen (2019), while keeping stratospheric winds nudged towards the model climatology. By doing so we are able to

remove the stratospheric variability and isolate the tropospheric pathway of ENSO to the North Atlantic, while allowing us to quantify the linearity of this pathway to the North Atlantic.

The paper is organized as follows: A description of the model simulations and the diagnostics employed is provided in section 2. In section 3, we describe the tropospheric pathway of ENSO to the North Atlantic by using experiments where the stratospheric winds are nudged towards climatology. Section 4 explores the spatial structure as well as the statistical robustness of the asymmetry and nonlinearity, while Section 5 focuses on the quantification of the nonlinearity as well as the relationship between the North Pacific and the North Atlantic circulation response to ENSO forcing. Finally, the propagation of waves in the troposphere from the North Pacific to the North Atlantic and how this mechanism contributes to the model NAO signal is shown in section 6. We close in section 7 with a brief summary and discussion of the main results.

## 2 Data and methods

### 2.1 Model description and experiments

In this study we use the Isca modelling framework (Vallis et al., 2018), which consists of the Geophysical Fluid Dynamics Laboratory (GFDL) dynamical core coupled with several configurable simplified physical parametrizations, including moist and radiative processes. Isca has been used to simulate atmospheric teleconnections by using SST forcing (e.g., Thomson and Vallis, 2018a, b; Jiménez-Esteve and Domeisen, 2019). In this study we use the same model configuration as in Jiménez-Esteve and Domeisen (2019), see the supplementary information therein for details about the model configuration. In the model, moist and radiative processes are considered through evaporation from the surface and fast condensation (i.e. no explicit liquid water content in the atmosphere), which interacts with the radiation and the convection scheme. We use the multi-band radiation scheme (rrtm) (Mlawer et al., 1997) used in the MiMA model (Jucker and Gerber, 2017), which allows configurable levels of ozone and $CO_2$ concentrations. We also use realistic topography and the continental outline from the ECMWF model (Dee et al., 2011). The land-sea contrast is achieved by changing surface characteristics such as the mixed layer depth, evaporative resistance and albedo as in Thomson and Vallis (2018a). SSTs are prescribed and thus we do not use a slab ocean or q-fluxes as in Thomson and Vallis (2018a, b). The model uses a Gaussian grid with a resolution of T42 and 50 vertical levels up to 0.02 hPa, of which 25 lie above 200 hPa.

The model sensitivity experiments consist of a climatological run and four experiments forced by linearly spaced magnitudes of tropical ENSO-like SSTs, i.e. strong and moderate EN and LN. In all experiments SSTs are globally prescribed and follow a repeating seasonal cycle (Figure 1b). In the climatological run global SSTs follow the 1958-2016 monthly SST climatology from NOAA ERSSTv4 (Huang et al., 2015) (daily values are linearly interpolated). The sensitivity experiments mimic canonical tropical Pacific ENSO-like SST anomalies, which consist of four identical spatial patterns (Figure 1a) with the same seasonal evolution but with a linearly changing magnitude between moderate and strong ENSO forcing of both signs (Figure 1b).

In the four ENSO experiments, climatological SSTs are prescribed north and south of 15 degrees and outside of the Pacific basin [150°E, 280°E] and only the positive (negative) parts of the SST anomalies are forced for EN (LN). Figure 1a displays

the December-January-February (DJF) mean SST anomalies for the strong EN forcing, and the three other forcings are multiples of it, with Niño3.4 region SST anomalies peaking slightly above $\pm 1.5(3.0)$K in November-December-January (NDJ) for moderate (strong) ENSO forcings (Figure 1b). This experiment design, despite being idealized, allows us to study the non-linearity/asymmetry in the atmospheric response arising solely due to changes in the ENSO magnitude, while removing the effect of the longitudinal location (central versus eastern Pacific ENSO events) and the asymmetry in the observed ENSO SST anomalies (EN events tend to be stronger than LN events).

With the objective to isolate the tropospheric from the stratospheric pathway, the five experiments use the same SST forcing as in Jiménez-Esteve and Domeisen (2019) (i.e. climatological, moderate/strong EN/LN), but the zonal mean winds in the stratosphere are relaxed towards the zonal mean seasonal cycle of the climatological SST simulation (Figure 1c shows the distribution averaged over Dec - Feb). This is achieved by applying a relaxation term of the form $(\overline{U} - \overline{U_{clim}})/\tau)$ to the prognostic equation for the zonal wind $U$ at all grid points, where $\overline{U}$ is the zonal mean of $U$ at a given time and $\overline{U_{clim}}$ is the zonal mean climatology target state (Figure 1c). $\tau = \tau(y, p, t)$ is the relaxation time given in days, which varies with pressure $p$, latitude $y$, and the month of the year $t$ as

$$\tau(y, p, t) = \begin{cases} \infty & \text{if } p > 0.5 p_{trop} \\ 5 + 15.83 \cdot (p/p_{trop} - 0.5) \, days & \text{if } 0.5 p_{trop} \leq p \leq 0.2 p_{trop} \\ 0.25 \, days & \text{if } p < 0.2 p_{trop} \end{cases} \tag{1}$$

where the tropopause pressure $p_{trop}(y, t)$ is computed for each latitude and month of the temperature climatology of the climatological simulation following the World Meteorological Organization definition (Słownik, 1992). Below $0.2 p_{trop}$ the relaxation time is 0.25 days, and at pressures higher than $0.5 p_{trop}$ the zonal winds evolve freely. Between $0.5 p_{trop}$ and $0.2 p_{trop}$ a linear function in pressure is applied in order to obtain a smooth transition between the nudged and the freely evolving atmosphere. The relaxation time distribution is shown in Figure 1d. We restrict the nudging to pressure levels above 0.5 times the tropopause level to avoid nudging winds within the upper part of the tropospheric jet, although no significant changes were observed when testing the sensitivity of the results to the position of the nudging zone. However, if the nudging is applied too close to the tropospheric jet, the variability in the troposphere strongly decreases and the response to the tropical ENSO forcing is damped.

In the nudged simulations employed in the present study, the interannual winter (DJF) NAO variance decreases by 40% with respect to the 5 identical simulations when the nudging is not applied, pointing to the important role of the stratosphere in North Atlantic variability. In the North Pacific, the effect of the nudging is much weaker than in the North Atlantic, and the Aleutian low variance decreases only by 10% when the nudging is applied. Additionally, the applied stratospheric nudging does not lead to any significant circulation anomalies in the tropospheric mean flow and the main modes of variability (EOF based) in the North Atlantic remain unaltered (Figure S3). In contrast, when we tested nudging the full climatological wind field instead of the zonal mean component this led to an undesired impact on the tropospheric variability and mean state, probably due to unrealistic changes in the planetary wave propagation.

All simulations are initialized from the same spun-up initial conditions, and are integrated for 80 years applying the strato-spheric nudging described above. An extra spin-up year is removed from the model data for each ENSO SST forcing simulation, which yields a total of 79 years for each ENSO SST forcing, and 80 years for the climatological SST simulation.

Unless indicated, the statistical significance of the EN and LN responses (forced minus climatological run) is assessed using a Monte Carlo approach using 1000 random DJF mean combinations of the climatological and forced simulations.

## 2.2 Dynamical diagnostics

### 2.2.1 Wave Activity Flux for Stationary Waves

The 3D WAF developed by Plumb (1985) is used here to describe the horizontal quasi-stationary (QS) Rossby wave energy propagation. This flux is phase-independent and parallel to the group velocity of the waves in the almost plane wave approximation. The horizontal components $(F_x, F_y)$ are computed as follows:

$$
\begin{pmatrix} F_x \\ F_y \end{pmatrix} = p\,cos\phi \begin{pmatrix} \frac{1}{2a^2cos^2\phi}\left[\left(\frac{\partial\psi^*}{\partial\lambda}\right)^2 - \psi^*\frac{\partial^2\psi^*}{\partial\lambda^2}\right] \\ \frac{1}{2a^2cos\phi}\left(\frac{\partial\psi^*}{\partial\lambda}\frac{\partial\psi^*}{\partial\phi} - \psi^*\frac{\partial^2\psi^*}{\partial\lambda\partial\phi}\right) \end{pmatrix}
\tag{2}
$$

where $a$ is the Earth's radius, $\lambda$ is longitude, $\phi$ is latitude, $p$ is the pressure level divided by $1000hPa$, and $\psi$ is the quasi-geostrophic stream function, calculated from the geopotential $\Phi$ using $\psi = \Phi/2\Omega sin\phi$, where $\Omega$ is the Earth's angular velocity. Asterisks indicate departures from the zonal mean.

To retain only the contribution from quasi-stationary (QS) waves, the daily means of the geopotential field are low-pass filtered with a cutoff period of 10 days prior to the calculation of the WAF. We only retain the contribution of zonal wavenumbers 1 to 3, since the large-scale planetary waves exhibit a stronger ENSO sensitivity (Jiménez-Esteve and Domeisen, 2018).

### 2.2.2 Wave Activity Flux for Transient Eddies

An equivalent WAF based on time deviations of the mean flow and independent of the phase speed allows the tracking of transient waves ($c \neq 0$) (Plumb, 1986). This formulation is independent of the phase speed, in contrast to the formulations of WAF for transient eddies developed by Takaya and Nakamura (1997, 2001), where the phase speed of the waves has to be inferred a priori. For a detailed formulation of the transient WAF and its climatology see Plumb (1986) and Nakamura et al. (2010, 2011). Atmospheric variables are decomposed into the transient part, denoted by $'$ and band-pass filtered with a period of 2 to 8 days, and its background flow mean part (computed using a 30 days low-pass filter), denoted by an overbar. The horizontal components of total transient WAF $(M_x, M_y)$ are computed as follows:

$$
\begin{pmatrix} M_x \\ M_y \end{pmatrix} = \frac{p\,cos\phi}{a|\nabla_h\bar{q}|} \begin{pmatrix} \frac{\partial\bar{q}}{cos\phi\partial\lambda}\overline{u'v'} + \frac{\partial\bar{q}}{\partial\phi}(\overline{v'^2} - \epsilon) \\ \frac{\partial\bar{q}}{cos\phi\partial\lambda}(\epsilon - \overline{u'^2}) - \frac{\partial\bar{q}}{\partial\phi}\overline{u'v'} \end{pmatrix} + \begin{pmatrix} \bar{u} \\ \bar{v} \end{pmatrix} M
\tag{3}
$$

where $M = \frac{1}{2} p cos\phi \overline{q' |\nabla_h \bar{q}|}$ is the quasi-geostrophic transient wave activity or pseudomomentum, and $q$ is the quasi-geostropic

potential vorticity. The first term on the right hand side in equation 3 is the so-called radiative part of the flux, where

$$\epsilon = \frac{1}{2} \left( \overline{u'^2} + \overline{v'^2} + \frac{Rp^\kappa}{H} \frac{\overline{\theta^{*2}}}{d\theta_0/dz} \right) \tag{4}$$

is the wave energy density, $\theta$ is the potential temperature, $R$ is the gas constant of air, $\kappa = R/c_p = 0.286$, $c_p$ is the specific heat at constant pressure, $H = 7km$ is the scale height. The parameter $\theta_0$ is the potential temperature averaged at each pressure level between $20°N$ and the pole.

### 2.2.3   Eady growth rate

The maximum Eady growth rate is calculated to study changes in baroclinicity, a proxy for baroclinic eddy development. It is computed as follows:

$$\sigma_E = 0.31 \frac{|f| \left| \frac{\partial u}{\partial z} \right|}{N} \tag{5}$$

(Vallis, 2013) where $N$ is the Brunt-Väisälä frequency ($N^2 = \frac{g}{\theta} \frac{\partial \theta}{\partial z}$), $g$ is the acceleration due to gravity, $\theta$ is the potential

temperature and $f$ is the Coriolis parameter.

## 3   Isolating the Tropospheric Pathway of ENSO to the North Atlantic

In this section, we investigate the winter (DJF mean) North Atlantic circulation response for the idealized ENSO SST forcings by relaxing the stratospheric zonal mean winds towards climatology. These experiments allow us to remove the ENSO remote influence through the winter polar stratosphere, i.e. the stratospheric pathway. In the model, the North Atlantic response peaks

in mid-winter (December to January) and does not exhibit significant changes in pattern, therefore the intra-seasonal variability of the response is not further investigated.

     Figure 2 shows the DJF mean sea level pressure (SLP) and the 250 hPa geopotential height (Z250) model response for the 4 different ENSO-like SST forcing simulations. The strongest ENSO response is observed in the North Pacific, where during EN conditions the AL intensifies (negative SLP anomalies) and extends eastwards (Fig2a,b). At upper levels (250hPa)

the EN response projects onto the positive PNA phase (Fig2e,f), i.e. the first mode of interannual variability in the North Pacific (Wallace and Gutzler, 1981). Qualitatively the opposite-signed pattern occurs for LN forcing, anomalies project onto the negative PNA phase, however significant asymmetry in terms of the strength and position of the North Pacific and North American anomalies can be identified. For example, the response for strong EN (Figure 2a) is stronger and covers a larger area than the opposite signed response for strong LN (Figure 2d and 5b). In this study, we focus on analyzing the response over

the North Atlantic sector. The reader is referred to Jiménez-Esteve and Domeisen (2019) for a study of the nonlinearity in the North Pacific region using the same model setup.

     In the North Atlantic, the model generally reproduces the observed ENSO-North Atlantic teleconnection (e.g., Jiménez-Esteve and Domeisen, 2018) and projects onto a negative NAO pattern (a decrease in the north-south SLP gradient) for the

strong EN forcing (Figure 2a), whereas the moderate EN forcing only leads to a significant response in the Northern lobe of the NAO, i.e. the Icelandic low, while having an insignificant impact on the Azores high (Figure 2b). However, note that the negative NAO dipole for strong EN does not extend into Europe, which is dominated by a positive SLP anomaly, which matches the observed response for the strongest EN event in 1998, which was characterized by the absence a SSW (see Figure 2 in Toniazzo and Scaife (2006)). The same SLP anomaly is obtained by Bell et al. (2009) using model experiments with a degraded stratosphere to remove the stratospheric pathway (see their Figure 11). Thus, although the strong EN forcing projects onto a negative NAO in the Atlantic, the impacts over Europe seem to be distinct. In contrast, the obtained SLP response for moderate EN differs from the study of Bell et al. (2009), which strongly resembles the SLP pattern obtained with the strong EN forcing. A likely explanation for this disagreement is the stronger SST forcing used in their study (Nino3.4>2K as compared to Nino3.4∼1.5 K in this study). The LN response generally exhibits the opposite-signed circulation anomalies, i.e. a deeper Icelandic low (IL), thus projecting onto the positive NAO phase (Figure 2c,d). For all forcings except for the strong EN, the impact over Europe is weak and insignificant, suggesting that the stratospheric pathway is an essential ingredient to fully describe the ENSO impact over Europe (e.g., Domeisen et al., 2015; Bell et al., 2009; Butler et al., 2014; Polvani et al., 2017; Trascasa-Castro et al., 2019; Cagnazzo and Manzini, 2009; Ineson and Scaife, 2009). The extratropical ENSO response exhibits a strong barotropic structure, i.e. Z250 and SLP anomalies exhibit a very weak westward tilt with height.

Thanks to the linearly varying strength of the ENSO SST forcing we can identify nonlinearities in the response. For example, in the model experiments we observe a saturation of the NAO response for LN: The Icelandic low SLP anomalies for moderate and strong LN are similar in strength, even though the forcing in the tropical Pacific SST is doubled in the latter case (Figure 2c-d). Another interesting result is that, except for the strong EN forcing, the surface response of the southern lobe of the NAO, i.e. the Azores high, is much weaker than the response of the Icelandic low.

We now analyze the associated changes in the tropospheric winds. Figure 3 displays the zonal wind response (shading) as well as the anomaly vectors at 850 hPa and 250 hPa for the four ENSO forcings. Consistent with the SLP and Z250 anomalies, the EN response is characterized by a strengthening and eastward extension of the Pacific jet, both at lower (Figure 3a,b) and upper levels (Figure 3e,f). This strengthening is not linear, i.e. the response is significantly stronger for strong EN than for moderate EN, consistent with the AL nonlinear response (Jiménez-Esteve and Domeisen, 2019). The meridional component of the winds is significantly more poleward along the western coast of North America during EN, advecting warm air to higher latitudes. Opposite-signed anomalies are observed for LN forcing, although weaker than for EN. Downstream in the North Atlantic, on average the jet stream weakens during EN while it slightly strengthens during LN, both at upper and lower levels. The Atlantic jet stream also becomes more tilted during EN (Figure 3e,f), while it is more zonal during LN (Figure 3g,h). Note that the model North Atlantic jet is too zonally oriented as compared to reanalysis. While near the surface (at 850 hPa) changes mainly correspond to a weakening (strengthening) of the zonal winds for EN (LN), at upper levels (250 hPa) the averaged response corresponds to a southward (northward) shift of the jet location, denoted by the north-south anomaly dipole of zonal wind. This weakening (strengthening) of the low level zonal winds for EN (LN) might be explained by a projection on the East Atlantic (EA) pattern (see Figure 11 in Woollings et al. (2010a)). The more southern location of the North Atlantic jet during EN has been also shown to be more thermally driven and less eddy driven (Madonna et al., 2019). Note that internally

generated NAO signals have been suggested to exhibit a more barotropic character than the observed ENSO response in the
North Atlantic (Mezzina et al., 2020).

The temperature at 850 hPa and the lower level baroclinicity in terms of the Eady growth rate (see methods) are shown
in Figure 4. Overall, temperature anomalies are stronger over land and are consistent with the changes in the 850 hPa wind
circulation (Figure 3a-d). Because SSTs outside of the tropical Pacific are kept fixed to climatological conditions for all exper-
iments, temperature anomalies at 850 hPa over the ocean are much weaker as compared to over land, where skin temperatures
are not fixed. The strongest 850 hPa temperature response is therefore located over the North American continent. For EN (LN)
experiments, this corresponds to higher (lower) than usual temperatures over Canada and lower (higher) temperatures over the
Southern US and Mexico. Thus, in general EN tends to weaken the meridional temperature gradient over North America,
whereas LN tends to strengthen it. This has an impact on the baroclinic zone east of the Rocky mountains and at the North
American Atlantic coast (Schemm et al., 2018), where the strong land-ocean temperature contrast enhances baroclinicity and
fuels the storm track.

Over Europe the strong EN forcing leads to warming (figure 4a), which is opposite to what should be expected from a
negative NAO, but consistent with the model positive geopotential anomalies over central Europe (Figure 2e) and increased
westerlies over Scandinavia (Figure 3a). The only other forcing leading to significant temperature anomalies over Europe is the
moderate LN (figure 4c), with a warming over Scandinavia due to strengthening of the westerly winds related to a deeper IL
(Figure 2c). Therefore the tropospheric pathway of ENSO exhibits significant asymmetry in the 850 hPa temperature response
to ENSO, with a weak warming both for EN and LN.

Over North America, EN related temperature anomalies weaken the baroclinicity from the central US to the Western Atlantic
while strengthening the baroclinicity from the eastern Pacific into the Gulf of Mexico (Figure 4e,f). Qualitatively the opposite
behaviour is observed for LN, however the magnitude of the Pacific response is much weaker and over North America the
strengthening and northward shift of the baroclinicity does not penetrate as far eastward in the North Atlantic as for the strong
EN forcing. A stronger (weaker) baroclinic zone along the North American Atlantic coast leads to a stronger (weaker) North
Atlantic storm track and therefore a positive (negative) NAO (Hoskins and Valdes, 1990), which is consistent with Figure
2a-d. For example, the weaker baroclinicity during strong EN tends to weaken the climatological Icelandic low, whereas the
strengthening of the meridional temperature gradient during LN can be linked to the intensification of the Icelandic low and
the associated near surface westerly winds (Figure 3c,d). Note that the baroclinicity response over North America is mostly
symmetric for moderate events (cp. Figures 4f,g), while it is more asymmetric for strong events (cp. Figures 4e,h).

# 4  Spatial pattern and statistical robustness of the nonlinear and asymmetric North Atlantic response to ENSO forcing

We now examine the spatial pattern of the nonlinear and asymmetric response to the ENSO tropospheric pathway (cp. Jiménez-
Esteve and Domeisen (2019)). We focus on SLP DJF averages over the North Atlantic (Figure 5), while the same analysis
for geopotential height at 250 hPa yields comparable results. We also do not investigate the intra-seasonal variations of the

nonlinear/asymmetric response as these are not significant in the simulations. First, we compute the asymmetry, i.e. the sum of the EN and LN responses, both for strong and moderate events (Figure 5a,b). The atmospheric response is symmetric if the same but opposite-signed response is found for EN and LN. The asymmetry of moderate events is then doubled to make it comparable to the strong events.

For moderate events (Figure 5a), the asymmetry pattern denotes a stronger AL/PNA impact for a moderate EN than for moderate LN, and a stronger positive NAO-like pattern for LN (compare Figures 2b,c). In the North Atlantic the wave train structure emerging from the Caribbean might be explained in terms of a stronger Rossby wave source (RWS) response in that area for moderate EN than for moderate LN (Figure S1) together with the asymmetric response of the transient WAF (section 6). Whereas in the North Pacific the asymmetry pattern is similar for strong and moderate events, in the Atlantic the asymmetry is quite different, and for strong events (Figure 5b) the asymmetry pattern arises due to a stronger impact on the southern lobe of the NAO for strong EN compared to LN, and the strong LN having an stronger impact on the IL region than strong EN (cp. Figure 2a,d). Positive asymmetries over Europe result as the distinct response for strong EN events (Figure 2a), although the origin of this response is not clear it might be related to the asymmetry in the tropical North Atlantic RWS response (Figure S1). Note that there are also significant differences between the asymmetry pattern in the North Atlantic shown in Figure 5a,b in (Jiménez-Esteve and Domeisen, 2019) and Figure 5a,b in this study. For example, the strongest negative NAO-like asymmetry in (Jiménez-Esteve and Domeisen, 2019) can be explained in terms of an asymmetry in the polar vortex response to ENSO in the model.

The nonlinearity within the EN and LN phase is computed by multiplying the response to moderate events by a factor of two and subtracting it from the strong event response. For an exactly linear response this would yield zero, as a doubling of the response would be expected in the linear case. However, for EN (Figure 5c) a zonal wave train pattern emerges. This nonlinearity pattern results from the superlinear deepening of the AL for the strong EN forcing in comparison to the moderate EN forcing, which is also located further eastward (denoted by a dipole structure, see also Jiménez-Esteve and Domeisen (2019)). In the North Atlantic, the negative SLP anomaly is suggestive of an eastward extension of the NAO-like dipole (cp. Figure 2a,b), but also of a Rossby wave train emerging around the Caribbean region that penetrates into Europe for strong EN events. This result is related to the finding of Hardiman et al. (2019) and Toniazzo and Scaife (2006), who showed that strong EN events exhibit a different response over Europe than moderate EN events, and that this response might be dominated by the tropospheric pathway through the Caribbean and assuming a saturation of the stratospheric pathway. However, the stratospheric pathway is not represented in our model experiments, and whether this pathway saturates is still debated as model studies do not agree. Note than in observations the strongest EN events were not accompanied by SSW events. Within the LN phase (Figure 5d) most of the nonlinearity is concentrated around the Icelandic low (IL) region, which denotes the saturation of the SLP response between the moderate and strong forcings (Figures 2c,d). At first this saturation of the IL response seems inconsistent with the linear increase in baroclinicity between moderate and strong LN (cp. Figure 4g,h), suggesting that other tropospheric mechanisms must account for the observed nonlinearity. In section 6 we show that the role of the WAFs may be responsible for this nonlinearity in the Icelandic low region for LN phase.

In order to quantify the statistical robustness of the asymmetric and nonlinear SLP response we employ a Monte Carlo technique following Garfinkel et al. (2018); Weinberger et al. (2019); Deser et al. (2018). The method consists in randomly selecting a sub-sample of increasing size from the pair of simulations used to compute the asymmetric or nonlinear component of the SLP response. This calculation is repeated 2000 times for the different randomly selected sub-samples in order to obtain a bootstrapped probability density function (PDF) of the respective variable. Successively increasing the size of the selected sub-samples allows us to answer the question of how many events must be considered before the nonlinearities/asymmetries become statistically detectable at a certain confidence level (in our case 95%) and for a certain region (see Figure 5 and section 5 for the exact definition of these areas).

Figure 6 displays the 95 and 50% confidence intervals of the asymmetry and nonlinearity of the SLP response in the different predefined regions (shown in different colors). For reference, Figure S2 displays the confidence intervals for each of the terms prior to compute the asymmetry and single phase nonlinearity. In Figure 6, when the whiskers, which indicate the 95% confidence interval, do not cross the zero-line the nonlinearity of that specific index becomes statistically detectable for a given number of events. For example the Aleutian low asymmetry between strong ENSO events becomes statistically detectable at 20 events, while 80 events are needed for the asymmetry in the Azores high index (Figure 6b). For reference Figure S2 displays the estimation of the PDFs for each of the terms used to compute the asymmetry and single phase nonlinearity.

A key message from Figure 6 should be that detecting nonlinearities in observations is still not possible, as a large amount of events is needed before these asymmetries/nonlinearities become statistically detectable. For example, to detect the asymmetry in the NAO between moderate EN and LN (Figure 6a and 5a) more than 50 events would be needed, and to detect the linearity within the LN phase (Figure 6d) in the Icelandic low region at least a sample size of 50 events is required.

In order to better understand how the ENSO signal reaches the North Atlantic, in the next section we investigate the relationship between the main modes of variability in the North Pacific and the North Atlantic, i.e. the AL and the NAO, respectively, and how ENSO might affect this connection.

## 5 The tropospheric link between the North Pacific and the North Atlantic variability

The relationship between the North Pacific and the North Atlantic atmospheric circulation is investigated using two indices based on SLP: The AL index is defined as the area-weighted average over [35-60N,180-240E] (green box in Figure 7a), and the NAO index is defined as the SLP difference between the Icelandic low (red) [50-75N,60-0W] and the Azores high (blue) [20-45N,60-0W] boxes (Figure 7b). Using Empirical Orthogonal Functions (EOF) to define the indices leads to similar results, showing that the model captures the main observed interannual variability in the two regions (see Figure S2 for the NAO pattern using EOF). We compute December to March monthly mean values for both indices and for each of the simulations. The two indices are standardized with respect to the climatological simulation. We use monthly instead of seasonal anomalies as these better represent the sub-seasonal timescales on which these pressure systems vary, but using seasonal means leads to comparable results.

The model monthly SLP anomalies regressed onto the December to March monthly AL and the NAO indices are shown in Figures 7a,b, respectively. As expected from its definition, the AL regression map has its main signal over the North Pacific and

350 corresponds to a strengthening/weakening of the AL climatological pressure system. In the same map, a north-south dipole over North America extending over the North Atlantic is also identified. This suggests that the negative (positive) NAO signature during EN (LN) can be achieved via the AL modulation and downstream influence, yet its influence does not significantly extend over Europe. The same conclusion is obtained when regressing the SLP onto the NAO index, in this case, apart from the expected North Atlantic dipole, which also extends into Europe, a monopole corresponding to the AL is also identified. One

interesting point here is that the NAO-like SLP dipole obtained when regressing SLP onto the AL index (Figure 7a) does not reach Europe as compared to the SLP regressed onto the NAO index (Figure 7b). This might be an indication that despite the AL having an influence on the NAO phase, other mechanisms like the stratospheric or the tropical Atlantic pathway might be needed to extend the ENSO NAO-like response over Europe.

The monthly probability density functions (PDFs) for the 5 model experiments using the different SST forcings are dis-

360 played in Figures 7c,d. These PDFs clearly show a strong ENSO influence on the AL index (Figure 7c). Color ticks on the x-axis indicate the composite mean anomalies in units of standard deviations. For the AL, there is a clear nonlinear and asymmetric response to the ENSO forcing, with a much stronger response for EN than for LN. The origin of this nonlinearity and asymmetry with respect to the ENSO SST forcing can be traced back to the tropical nonlinear relationship between the convective upper troposphere divergent wind response and the underlying tropical SST anomalies (e.g. Johnson and Kosaka,

2016). The North Pacific surface circulation then reacts to the divergence in a linear fashion (see Figure 4c in Jiménez-Esteve and Domeisen (2019)).

The ENSO impact projecting onto the NAO pattern via the tropospheric pathway is much weaker than on the ENSO impact on AL, as has already been shown in the previous section. In general, during winter EN (LN) the tropospheric pathway projects onto a negative (positive) NAO-like pattern (Figure 7d). Strong EN forcing tends to produce a stronger response than the

370 opposite-signed LN. Consistent with Figures 2b,c the moderate LN forcing projects more strongly onto the positive NAO than the moderate EN projects onto a negative NAO. In fact, the moderate EN response closely resembles a blocking pattern (Figure 2b) which projects onto the second most important mode of variability in the model (Figure S3). This figure also shows a saturation of the NAO response for LN forcing, with a mean NAO response around -0.4 standard deviations for both moderate and strong LN, that is, although these are separated by a doubling in the SST forcing. A possible explanation for

these nonlinearities in the tropospheric pathway is explored in terms of eastward WAFs in section 6.

While the PDF of the AL is overall symmetric, the PDF of the NAO is negatively skewed, i.e. it has a longer tail towards negative NAO values, which is in agreement with observations (e.g., Woollings et al., 2010b; Domeisen et al., 2018). We find that EN (LN) tends to increase (decrease) the standard deviation of both the AL and the NAO (Figure S4) and that the strong EN forcing acts to decrease the negative skewness of the climatological NAO making its pdf more symmetric, similar to the

380 AL (Figures7c,d and S4). Despite this decrease in the negative skewness, there is still a significant increase in the occurrence of extreme negative NAO events during the strong EN forcing. Note that we cannot compare this figure with reanalysis as there

have not been a sufficient number of strong ENSO events in the observational record to calculate the corresponding PDF for the NAO.

Figure 8 shows the DJF seasonal mean of the NAO in terms of the AL index for the five simulations with a nudged strato-
sphere. The correlation coefficient between the winter AL and the NAO indices is significant and larger than 0.5, which is much
larger than in reanalysis (Figure S5). Actually, the weaker signal in reanalysis might be related to the destructive interference
of other sources of variability and the fact that the three strongest EN events were also accompanied by a strong polar vortex.
Figure 8 also illustrates how the large internal variability in the extratropics can mask the North Pacific influence on the NAO
variability in response to ENSO. For example, positive and negative values of the NAO index occur for any of the ENSO
forcings, as also shown in Figure 11 in Trascasa-Castro et al. (2019). Using obserations

Despite the large extratropical variability, and the small signal-to-noise ratio, the relationship between the North Pacific and
North Atlantic ENSO response seems is linear to a good approximation, thus it seems that most of the modeled asymmetry
between the strong EN and strong LN (Figure 5b) forcing projecting onto the NAO pattern should mainly originate from the
asymmetry in the tropical Pacific upper level divergent wind response (see Figure 4a in Jiménez-Esteve and Domeisen (2019)).

## 6 Wave activity fluxes of transient and quasi-stationary waves in response to ENSO forcing

The connection between the North Pacific and North Atlantic in the troposphere is predominantly driven by the downstream
propagation of QS and transient waves (e.g., Li and Lau, 2012a; Jiménez-Esteve and Domeisen, 2018; Schemm et al., 2018).
In this section, we analyze the modeled tropospheric circulation anomalies associated with increased eastward propagation of
these waves. Due to the model experiment design, our results isolate the tropospheric pathway from stratospheric interaction,
which could otherwise exert an influence on the propagation of waves within the troposphere (e.g., Castanheira and Graf, 2003;
Sun and Tan, 2013; Gong et al., 2019).

Figure 9 displays SLP and Z250 composites with respect to strong eastward transient and QS WAF monthly anomalies across
North America. Details about the calculation of the WAF are provided in the methods section. The monthly mean eastward
component of the transient WAF ($M_x$) at 250hPa is averaged over the area [20-40N,220-300E] (green box in Figure 9a), which
is the climatological location where most of the baroclinic transient eddies propagate (Nakamura et al., 2010). Equivalently,
the monthly mean eastward component of the large-scale (k=1-3) QS WAF is averaged over a more northern location [45-
65N,220-300E] (green box in Figure 9e), i.e. the preferred climatological location of the strongest eastward planetary QS WAF.
December to March monthly means of the two indices are standardized with respect to the climatological SST simulation. For
the composites we choose a 1.5 standard deviation threshold, but using other thresholds leads to qualitatively similar results.

When the eastward transient WAF is increased over the southern US (green box in Figure 9a) the composite mean AL
is stronger than climatology and the North Atlantic anomalies project onto a negative NAO. Thus, our model experiments
reproduce well the SLP response to increased eastward transient WAF observed in reanalysis (cp. Figure 3 in Jiménez-Esteve
and Domeisen (2018)). At upper levels (Figure 9b) this teleconnection corresponds to a positive PNA-like Rossby wave train,
which coincides with a decrease in the eastward large-scale QS WAF.

The sensitivity to the different ENSO SST forcings is shown as probability distribution functions in Figure 9c, while table 1 shows the percentage of the strong positive (above 1.5 standard deviations) and negative (below -1.5 standard deviations) monthly eastward transient and QS WAF events. According to Figure 9c, in the LN simulations the eastward propagation of transient eddies along the southern part of the US is clearly reduced, with very few events above the 1.5 standard deviations threshold, i.e. 0.3% for both the moderate (strong) LN forcing, respectively (Table 1). In contrast, the eastward transient WAF distribution is shifted to positive values for the strong EN forcing, while no sensitivity is observed for the moderate EN forcing. Strong transient WAF events occur on average for 13% of the months for the strong EN forcing.

An increased eastward QS WAF coincides with a weakening of AL and negative Z250/SLP over north-eastern Canada and Greenland, which projects onto the negative PNA phase and the positive NAO phase (Figure 9d,e). This response is strongly barotropic, which supports the fact that the circulation anomalies are indeed forced by QS waves. While for the increased propagation of transient eddies there is a robust decrease of the QS WAF over Canada (Figure 9b), the opposite is not true. During increased eastward QS WAF events, the westward transient WAF anomalies do not penetrate into the southern NAO region (Figure 9d), and thus there is a weaker impact there.

Because the weakening of the AL is more likely to occur during LN winters (Figure 7c), the probability of increasing the eastward QS WAF (favoring a positive NAO) is larger during LN. This is supported by Figure 9f, which shows a positive shift of the PDF to a more eastward QS WAF during LN. Yet, this response is stronger for the moderate LN than for the strong LN forcing. This is confirmed for the extreme QS WAF events, with a 15.5(9)% frequency for the moderate (strong) LN forcings (Table 1). The opposite QS WAF response is observed for EN experiments, however here the response is more linear as the strongest decrease is observed for the strong EN forcing.

Figure 10 shows the NAO index dependence on the transient and the QS eastward WAF indices. For this figure all five simulations with a nudged stratosphere have been used and monthly averages from December to March are used to better represent the low frequency variability of the NAO. The averaged values of the NAO are distributed into 2-dimensional bins of 0.6×0.6 standard deviations with respect to the transient and QS WAF standardized indices. This representation allows us to differentiate between the individual and combined effects of the transient and the QS waves propagating downstream from the North Pacific to the North Atlantic.

Due to the opposite-signed response to EN and LN, transient and QS WAFs exhibit a correlation, i.e. low (high) values of eastward transient WAF tend to simultaneously occur with high (low) values of QS WAF. Therefore, the NAO response to ENSO results from a constructive interference between these two types of waves (upper-left and lower-right corners in Figure 10). Destructive interference between these two mechanisms can occasionally occur (lower-left and upper-right corner), however these events are less frequent. One possible way to interpret the ENSO impact on the NAO is via the changes in 2-dimensional distribution of these two WAF indices. The 2-dimensional PDFs for each ENSO forcing are represented by the colored lines in Figure 10. The saturation of the NAO response during LN (Figure 7d) is consistent with the strong LN forcing leading to weaker transient WAF than the moderate LN, which is compensated by the stronger QS WAF response. Figure 7 also explains why the moderate EN forcing projects weakly onto the negative NAO phase, as neither the transient nor the QS WAF distribution is significantly shifted from the climatological values.

## 7 Summary and Discussion

Idealized atmospheric model experiments with seasonally evolving prescribed SSTs have been conducted to explore the tropospheric pathway of ENSO to the North Atlantic, as well as potential nonlinearities in this pathway. The model configuration allows us to remove potential nonlinearities arising from the observed asymmetry in the magnitude and location of LN and EN SST patterns, as well as from extratropical SST effects. To isolate the tropospheric from the stratospheric pathway we use simulations where stratospheric winds are nudged towards the model zonal mean climatology. Our results can be summarized as follows:

1. Without a stratospheric influence on the troposphere, the North Atlantic atmospheric response to ENSO forcing can be explained in terms of the upstream influence from the North Pacific. The response to ENSO in the North Atlantic projects onto a negative (positive) NAO during EN (LN). However, only the strong EN forcing reproduces a complete negative NAO dipole, whereas the other ENSO forcings exhibit a stronger impact on the Icelandic low.

2. The ENSO tropospheric pathway to the North Atlantic exhibits significant nonlinearity and asymmetry with respect to the tropical Pacific SST forcing, both in terms of the location and the strength of the impacts, although more than 40-50 events are required before these are statistically detectable. Strong EN forcing has a stronger impact on the NAO than strong LN forcing, but moderate LN forcing has a significant stronger impact than moderate EN forcing. For LN forcing, there is a saturation of the positive NAO response with no further increase in the NAO index even when doubling the SST forcing (Figure 7d). Such a saturation effect is not observed for El Niño.

3. The Aleutian low and the NAO modes of variability are significantly correlated at monthly and seasonal timescales (Figure 8) through tropospheric dynamics only.

4. Consistent with reanalysis (Jiménez-Esteve and Domeisen, 2018), in the model EN forcing increases (decreases) the eastward wave activity flux (WAF) of transient eddies (large-scale QS waves) across North America. Because these two types of WAF exhibit an opposite response for EN and LN (Figure 9), in the model the NAO response to ENSO results from a constructive interference between the impacts of the two WAFs (Figure 10). The asymmetry in eastward WAF response for moderate EN and LN (Figure 9) together with larger Rossby wave source anomalies for moderate EN, likely explain the stronger NAO projection of moderate LN than moderate EN forcing (Figures 5a and 6a).

While this study has focused on isolating the tropospheric pathway of ENSO to the North Atlantic, in the real world the stratosphere can play an important role in communicating the ENSO signal to the North Atlantic (e.g., Butler et al., 2014; Domeisen et al., 2015).

In particular, the role of the stratosphere in contributing to nonlinearities is still debated: When asymmetries between EN and LN in the tropical Pacific are removed, the stratospheric response is significantly more asymmetric (Trascasa-Castro et al., 2019) than if forcing asymmetries are not removed (Weinberger et al., 2019) in different atmospheric models, while Rao and Ren (2016a, b) using a model and reanalysis showed a nonlinear and asymmetric stratospheric wind response. In their

experiments, Trascasa-Castro et al. (2019) also find that the NAO response to EN is approximately linear and does not saturate with increasing forcing (up to 3.0 K in DJF),whereas for LN this response is weak and asymmetric with respect to the EN response.

Different modeling studies also find differences in the relative importance of the stratospheric and tropospheric pathways for LN. While Hardiman et al. (2019) using an ensemble of seasonal hindcasts find that the stratospheric pathway to the North Atlantic dominates over the tropospheric pathway for strong LN, i.e. a significant strengthening of the polar vortex; Trascasa-Castro et al. (2019) find a very weak stratospheric response for LN in mid-winter, and thus a insignificant NAO response.

Several factors might be able to explain the differences among the above studies: e.g. differences in the model and simulation 490    setup, the background SSTs on which the ENSO forcing is imposed (e.g., Xie et al., 2018), the role of the extratropical SST anomalies, the location and pattern of the ENSO SST forcing, the location and intensity of the climatological planetary-scale stationary waves, and the stratospheric variability of the model. The too zonally oriented North Atlantic jet bias present in many GCMs (Zappa et al., 2013) might also affect the ability of the model to simulate ENSO teleconnections. Further work will be necessary to assess the model dependence in relation to the ENSO stratospheric pathway. In the present study we 495    find that nonlinearities in the North Atlantic atmospheric response to ENSO can originate within the tropospheric pathway, independently of the stratospheric response to ENSO.

Another important factor that can impact the ENSO teleconnection to the North Atlantic is decadal variability such as the Pacific Decadal Oscillation (PDO) (Rao et al., 2019) and the Atlantic Mutidecadal Oscillation (AMO) (Zhang et al., 2019). The ENSO-stratospheric teleconnection has weakened in recent decades (Hu et al., 2017; Domeisen et al., 2019), which the 500    PDO variability alone cannot explain (Rao et al., 2019), but circulation anomalies over Eastern Europe have likely contributed (Garfinkel et al., 2019). The present study characterizes the North Atlantic response to ENSO and its linearity during these periods when the stratospheric pathway is inactive.

Furthermore, the influence of other sources of interannual variability like the Quasi-Biennial Oscillation (QBO) (e.g., Calvo et al., 2009; Garfinkel and Hartmann, 2010; Hansen et al., 2016), the Madden-Julian Oscillation (MJO) (e.g., Hoell et al., 2014) 505    or Tropical Atlantic SST anomalies (e.g., Toniazzo and Scaife, 2006; Sung et al., 2013; Rodríguez-Fonseca et al., 2016) can modulate the ENSO signal to the North Atlantic. These modes of variability are not present in our simplified model and we can therefore exclude their influence. In the model simulations of Hardiman et al. (2019) the strong EN response in early-to-mid winter is characterized by a Rossby wave source in the Caribbean and tropical Atlantic, with linearly varying strength for symmetric EN and LN events. Our strong EN forcing is characterized by a positive SLP response over Europe, which closely 510    resembles the results obtained by Bell et al. (2009) using simulations with a degraded stratosphere and prescribed SSTs. Thus, although the full wave-like pattern obtained by Hardiman et al. (2019) seems to require SST anomalies outside of the tropical Pacific (Rodríguez-Fonseca et al., 2016), a Caribbean Rossby wave source might be able to explain the different response for strong EN events over Europe in our experiments. This mechanism might be relevant for strong EN events as suggested by Toniazzo and Scaife (2006) assuming a saturation of the stratospheric pathway, or during early winter when the North Pacific 515    pathway and the stratospheric pathway are not yet fully developed (Ayarzagüena et al., 2018).

The location of the maximum SST anomalies in the tropical Pacific, i.e. the ENSO flavor, can further influence ENSO teleconnection. The sensitivity to ENSO flavor has been excluded in this study through the design of the model experiments. Using reanalysis and CIMP5 models, Iza and Calvo (2015); Calvo et al. (2017) found a weaker polar vortex during eastern Pacific EN and no significant anomalies during central Pacific EN. However, despite the weaker impact on the polar vortex,
Graf and Zanchettin (2012) found that the tropospheric pathway is stronger for central Pacific EN events, leading to a stronger negative NAO impact in the Atlantic and extending its influence further into Europe. These differences can be explained in terms of the strength and location of the AL, which is itself linked to the intensity and longitudinal location of the convective response in the tropical Pacific (Garfinkel et al., 2018). The variability in the location of the tropical SST forcing can also contribute to nonlinearity in the extratropical winter teleconnections (e.g., Yu et al., 2012; Frauen et al., 2014). Thus, further
research will help to understand the nonlinearity in the NAO response arising from the location of the Pacific SST forcing.

In summary, our model experiments confirm most of results obtained using reanalysis (Jiménez-Esteve and Domeisen, 2018), while providing further insight into the nonlinearity and asymmetry of the tropospheric pathway of ENSO, which might become more relevant for seasonal prediction in the North Atlantic and Europe due to the observed weakening of the stratospheric pathway in recent decades.

*Code and data availability.* The ISCA modelling framework was downloaded from the Github repository (https://github.com/ExeClim/Isca). The ERSSTv4 data was downloaded from the NCAR research data archive (https://rda.ucar.edu/) and the ERA-Interim reanalysis from the ECMWF server (https://apps.ecmwf.int/datasets/data/).

*Author contributions.* Bernat Jiménez-Esteve performed the model simulations, the data analysis and plotting, and wrote the first draft of the manuscript. Daniela I.V. Domeisen significantly contributed to the interpretation of the results and the writing of the paper.

*Competing interests.* The authors declare that they have no conflict of interest

*Acknowledgements.* The authors would like to thank Stephen Thomson and Edwin Gerber for their helpful comments regarding the model setup. We are also grateful to Blanca Ayarzagüena and Chaim Garfinkel for constructive discussions. Support from the Swiss National Science Foundation through project PP00P2_170523 is gratefully acknowledged. The authors would like to thank Paloma Trascasa-Castro and two anonymous reviewers for their time and insightful comments during the review of our manuscript.

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

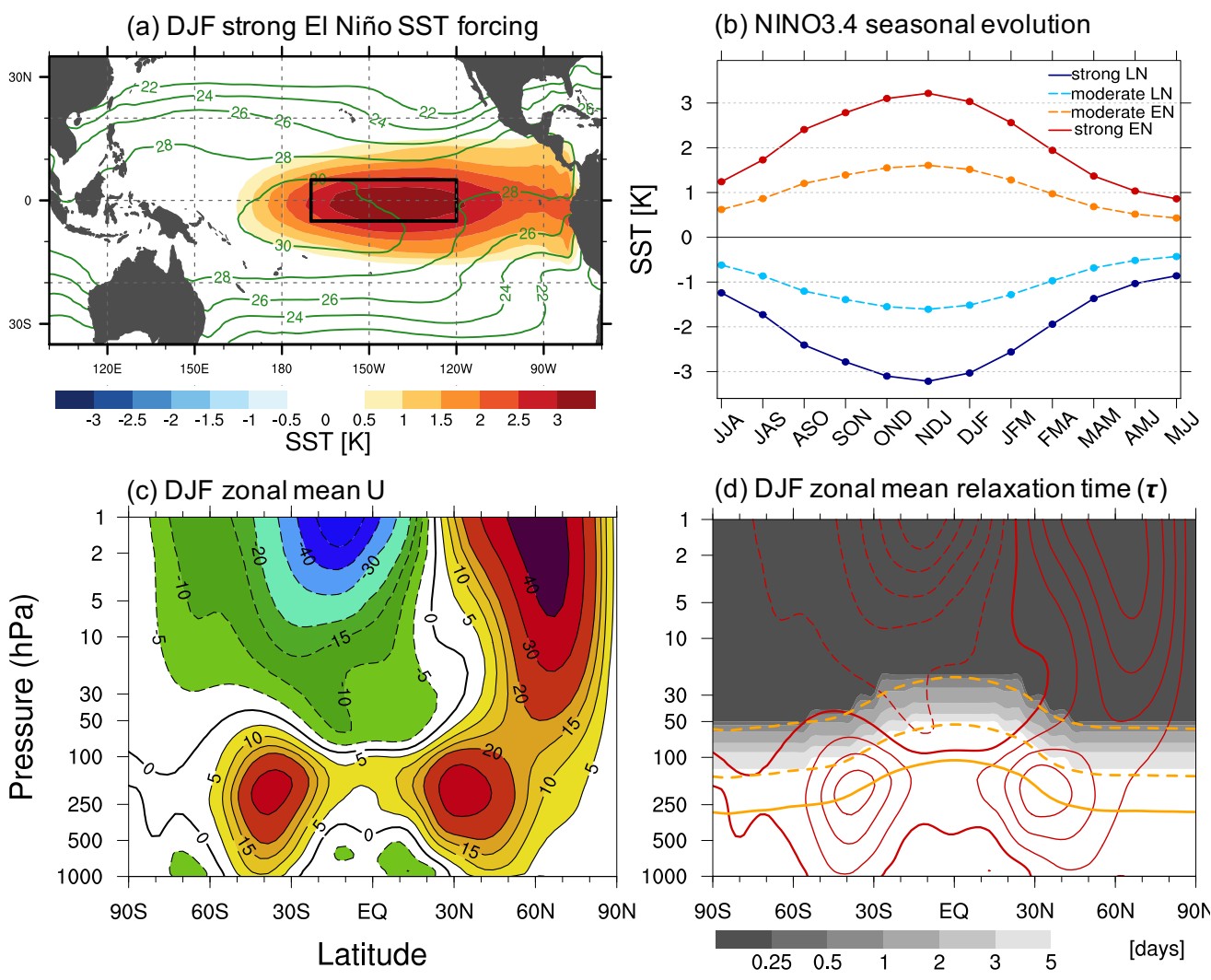

**Figure 1.** (a) DJF SST anomaly pattern in the tropical Pacific for the strong El Niño forcing simulation. (b) The seasonal evolution of the SST anomalies in the NINO3.4 region (black box in (a)) for the four types of SST ENSO forcings. (c) The DJF zonal mean zonal wind (U) in the climatological SST simulation. (d) The DJF zonal mean relaxation timescale ($\tau$) in days. In (d), red contours represent U as in (c), the solid orange line represents the DJF mean tropopause height and the dashed orange lines represent the $0.5p_{trop}$ and $0.2p_{trop}$ levels, which denote the limits for the transition area of the nudging.

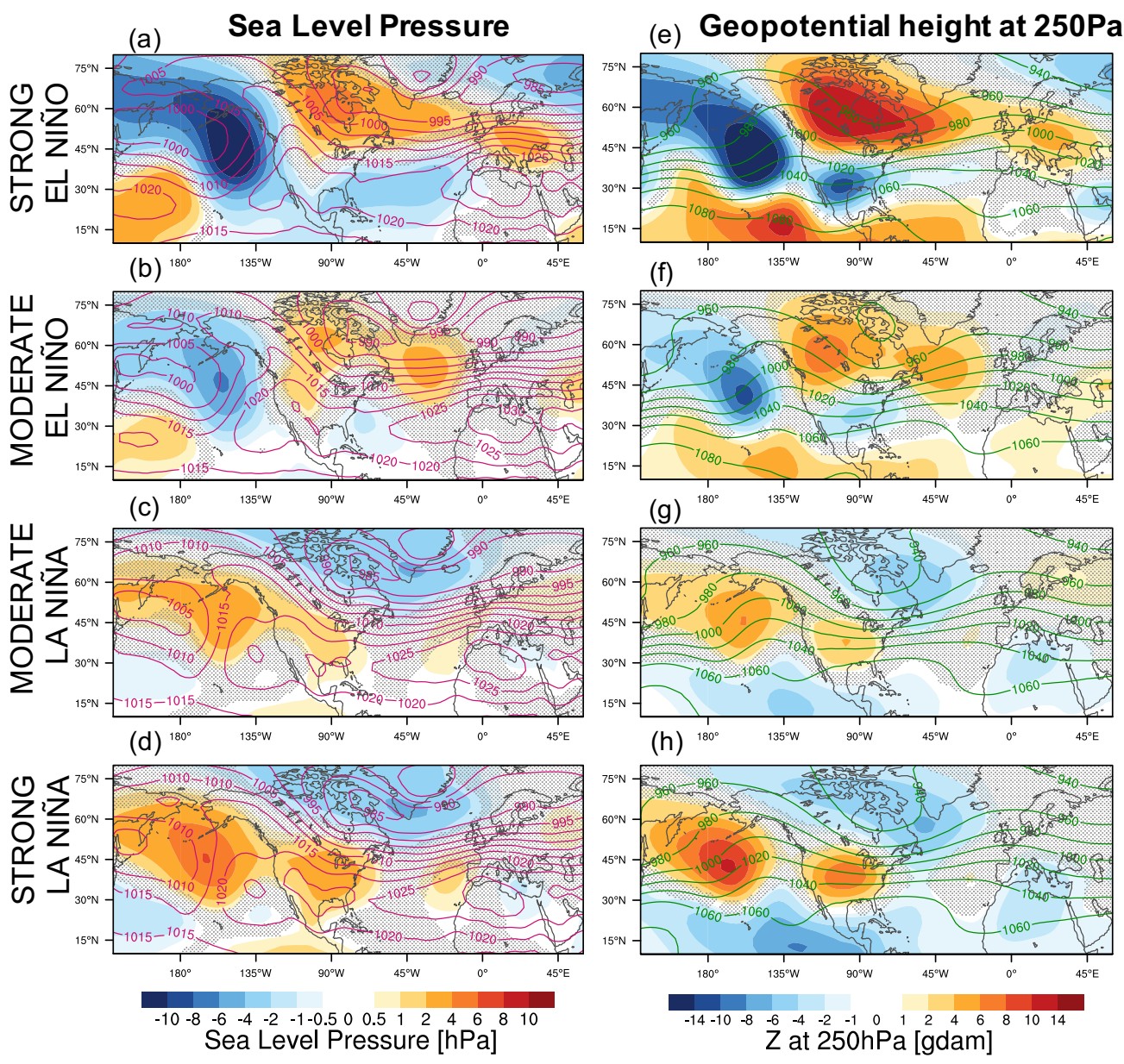

**Figure 2.** (a-d) DJF SLP and (e-h) DJF mean Z250 model response for the nudged stratosphere simulations with (a,e) strong EN, (b,f) moderate EN, (c,g) moderate LN, and (d,h) strong LN forcing. Contour lines indicate absolute values of SLP (hPa) and Z250 (gdam = geopotential decameters), respectively. Non-statistically significant values below the 95% confidence level are dotted in grey.

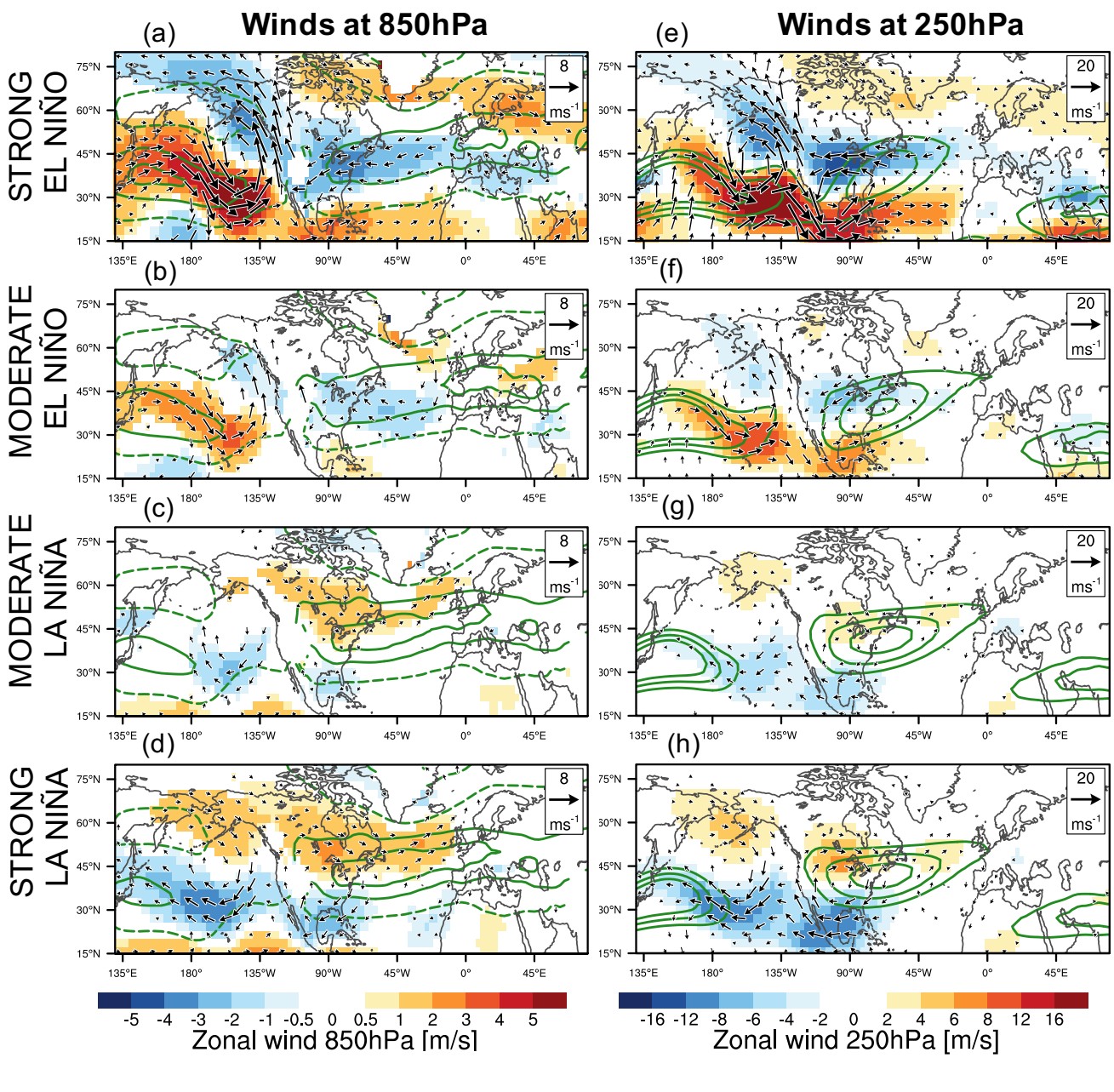

**Figure 3.** As in Figure 2, but for the zonal wind (a-d) at 850 hPa and (e-h) at 250 hPa. Arrows display the zonal and meridional wind anomalies. Green contour lines show the absolute values of the zonal wind component for each simulation [0 (dashed), 5 and 15 m/s in (a-d) and 30, 40 and 50 m/s in (e-h), respectively]. Only statistically significant values above the 95% level are shown.

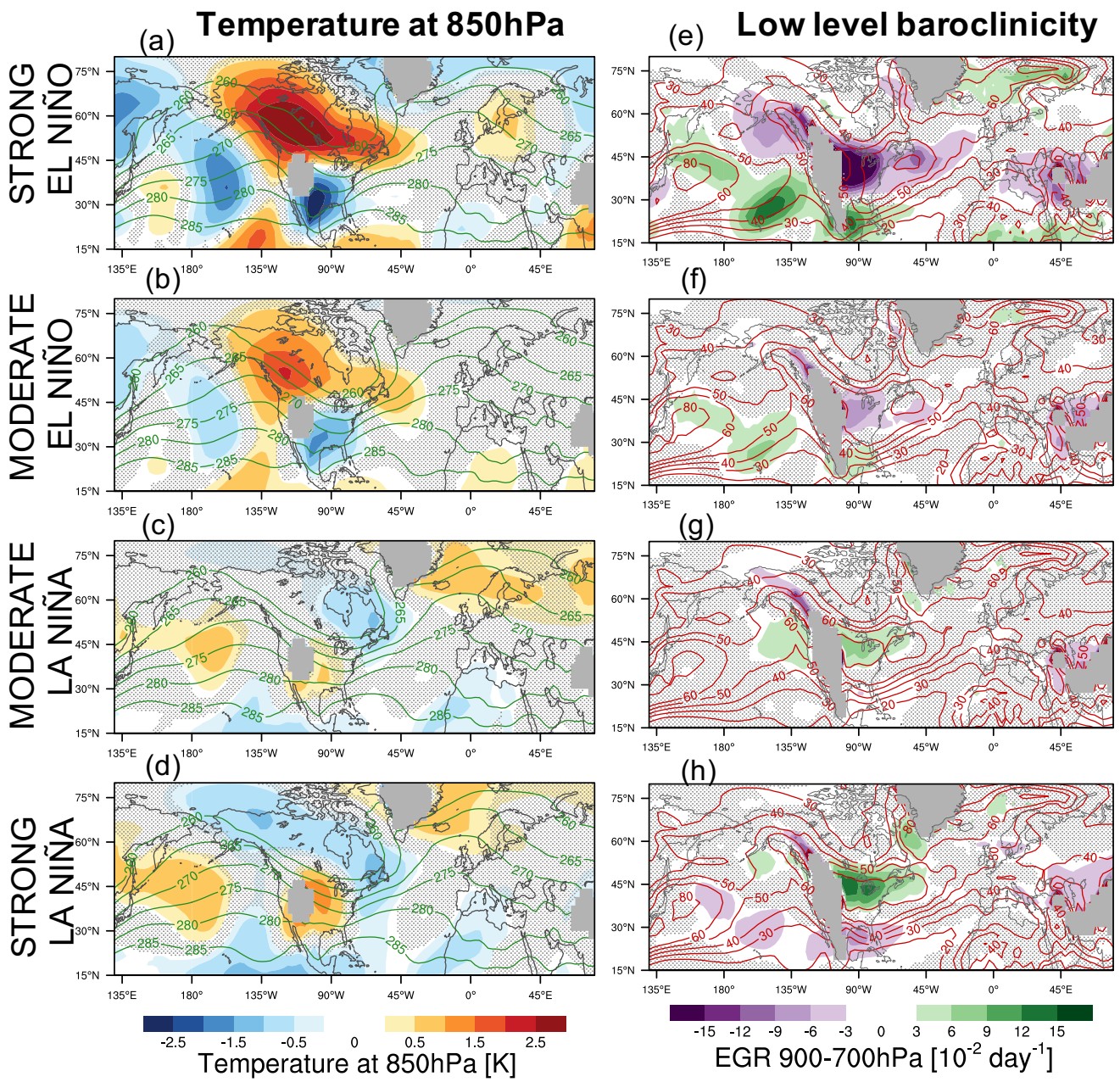

**Figure 4.** As in Figure 2, but for (a-d) the temperature at 850 hPa and (e-h) the Eady growth rate (EGR) vertically integrated between 900 and 700 hPa. Contour lines indicate the absolute values of temperature (K) and EGR ($10^{-2}$ day$^{-1}$), respectively. Non-statistically significant values below the 95% confidence level are dotted in grey.

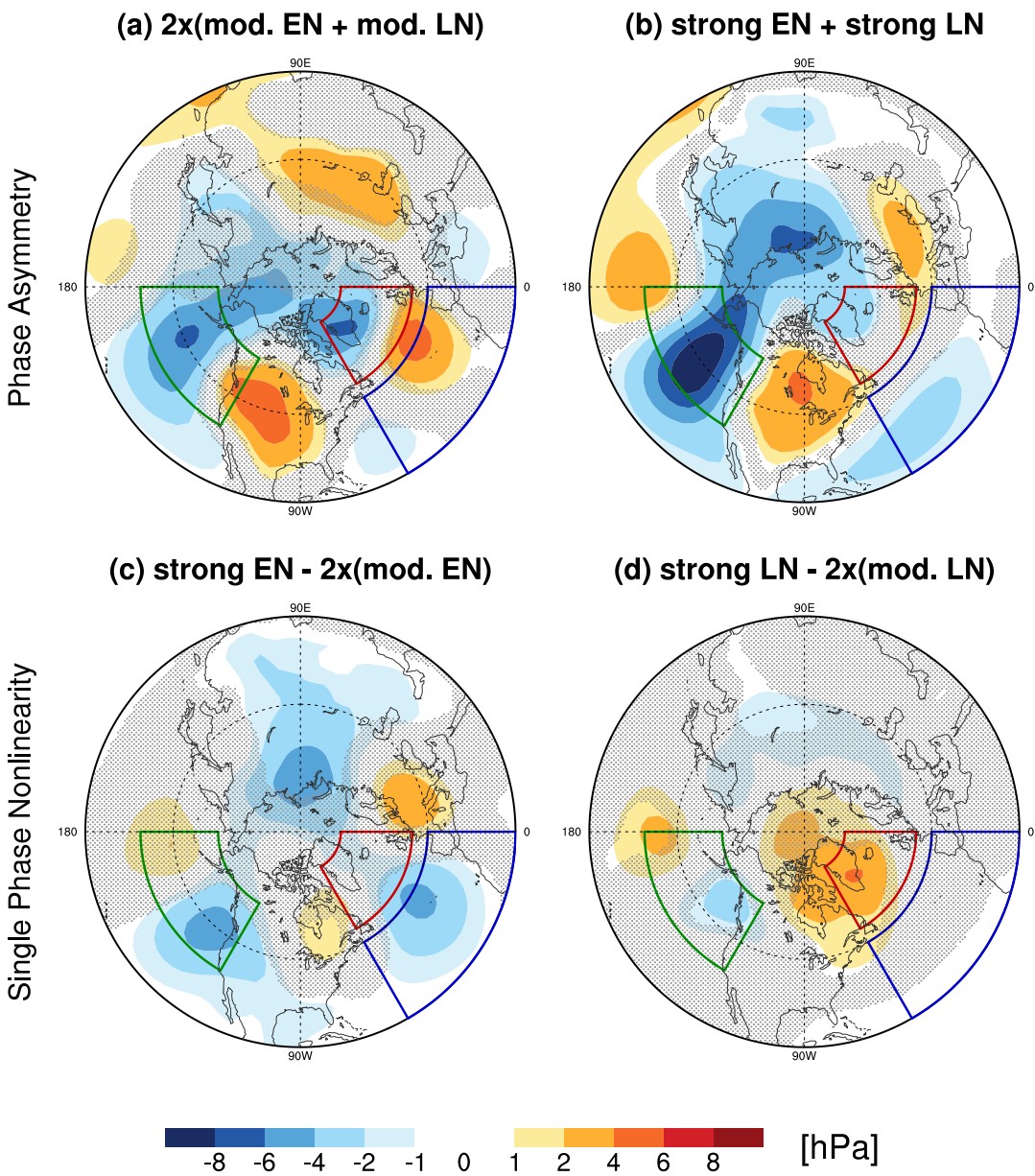

**Figure 5.** DJF model SLP response asymmetry for (a) twice the moderate and (b) strong ENSO forcings. (c) EN and (d) LN single phase nonlinearity. See the main text for definitions. Green, red and blue boxes indicate the regions used to define the AL, the Icelandic low and Azores high regions, respectively, see main text for details. The NAO is computed as the difference between the Icelandic low and the Azores high index. Non-statistically significant values below the 95% confidence level are dotted in grey.

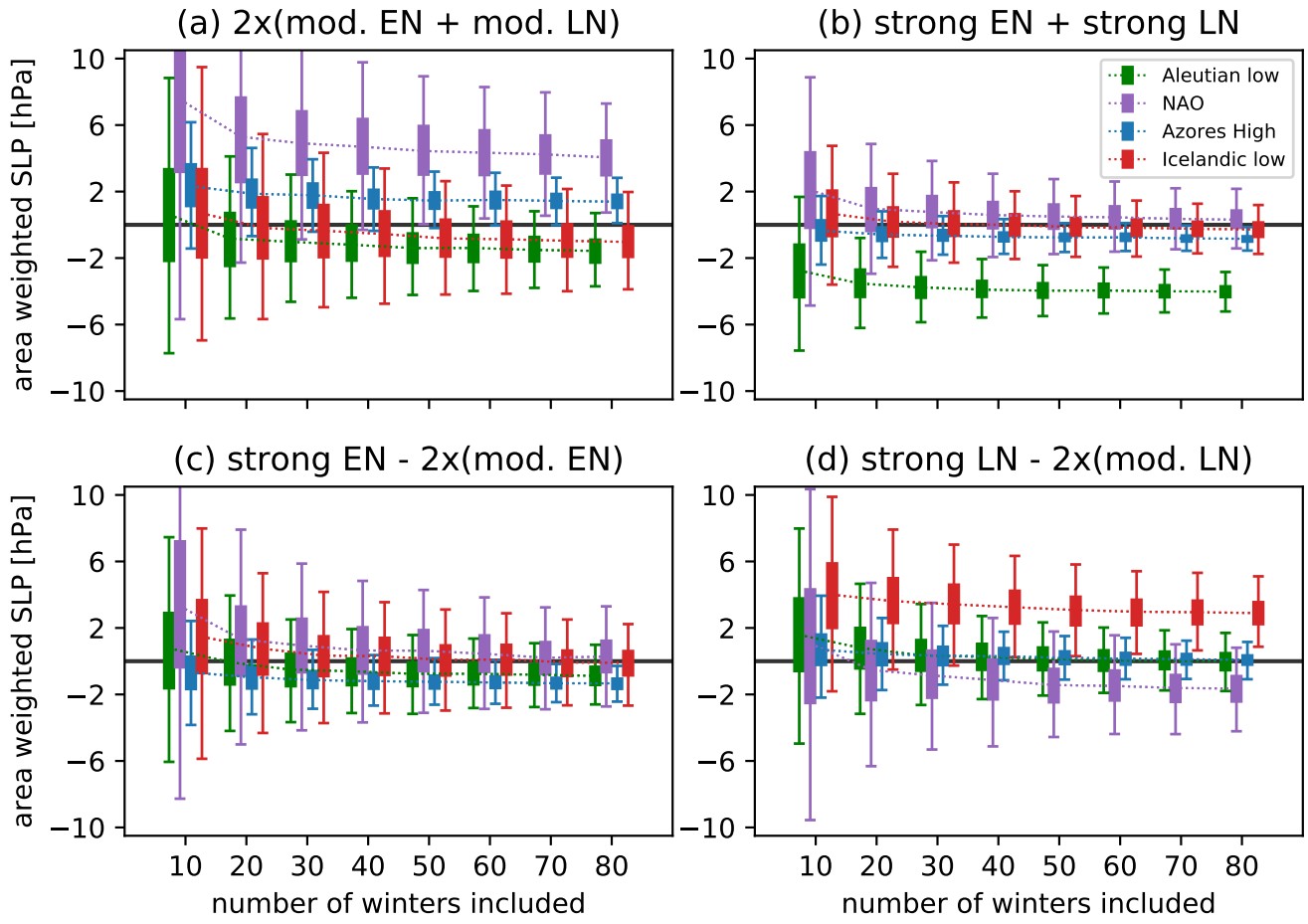

**Figure 6.** Box plot displaying the 95(50)% confidence intervals indicated by the whiskers (solid boxes) of the DJF model SLP asymmetric response for (a) twice the moderate and (b) strong ENSO forcings when the winter anomalies in these experiments are randomly sub-sampled in groups of increasing size (shown on the x-axis). Colors indicate the different SLP indices (green: Aleutian low, purple: NAO, blue: Azores High, red: Icelandic low). (c) the same as (a,b) but for EN and (d) LN single phase nonlinearity. When the whiskers do not touch the zero-line for a specific sample size and magnitude, then the asymmetry/nonlinearity is statistically detectable at the 95% confidence interval.

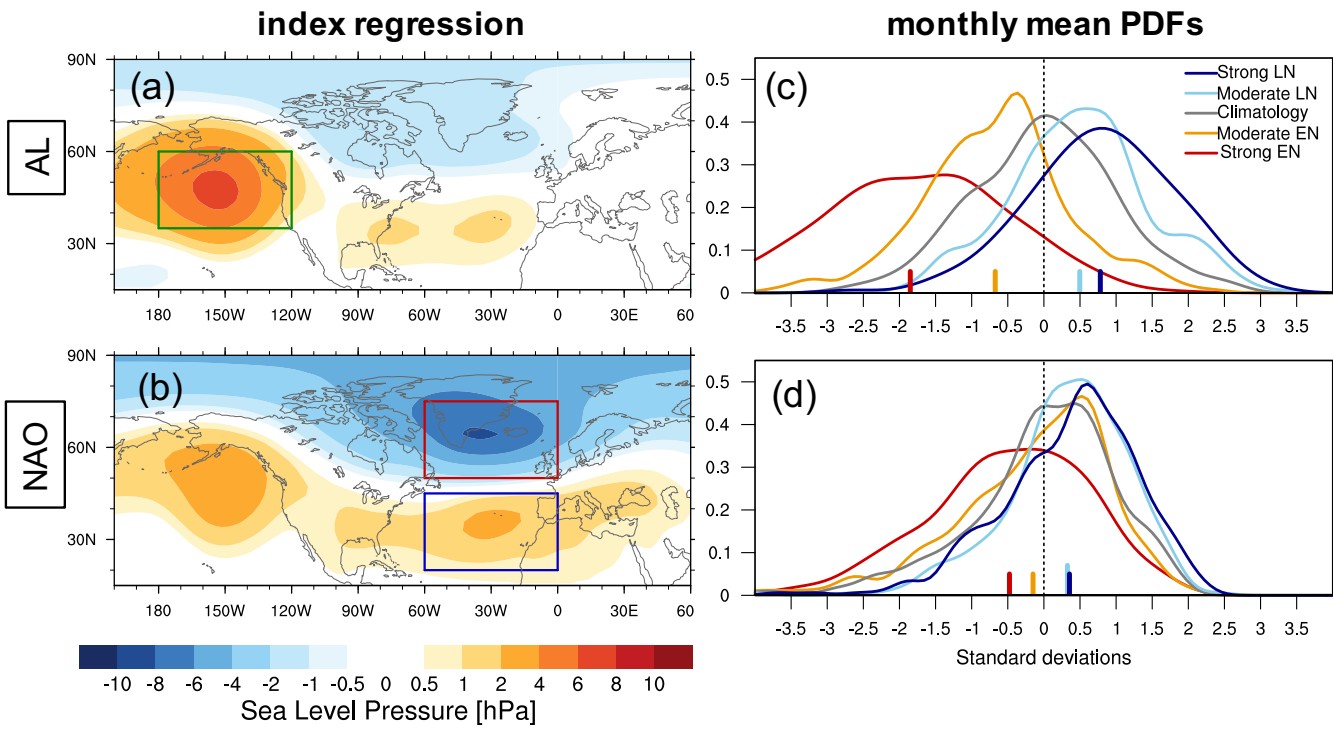

**Figure 7.** December to March monthly SLP (in hPa) regressed onto (a) the AL and (b) the NAO indices, using the five simulations with stratospheric nudging (396 years). Green, red and blue boxes indicate the regions used to define the AL and NAO indices, respectively, see main text for details. The PDFs of the December-January-February-March standardized monthly means for (c) the AL and (d) the NAO indices. Colors represent the PDFs for the different ENSO forcings (red: strong EN, orange: moderate EN, grey: climatology, light blue: moderate LN, dark blue: strong LN).

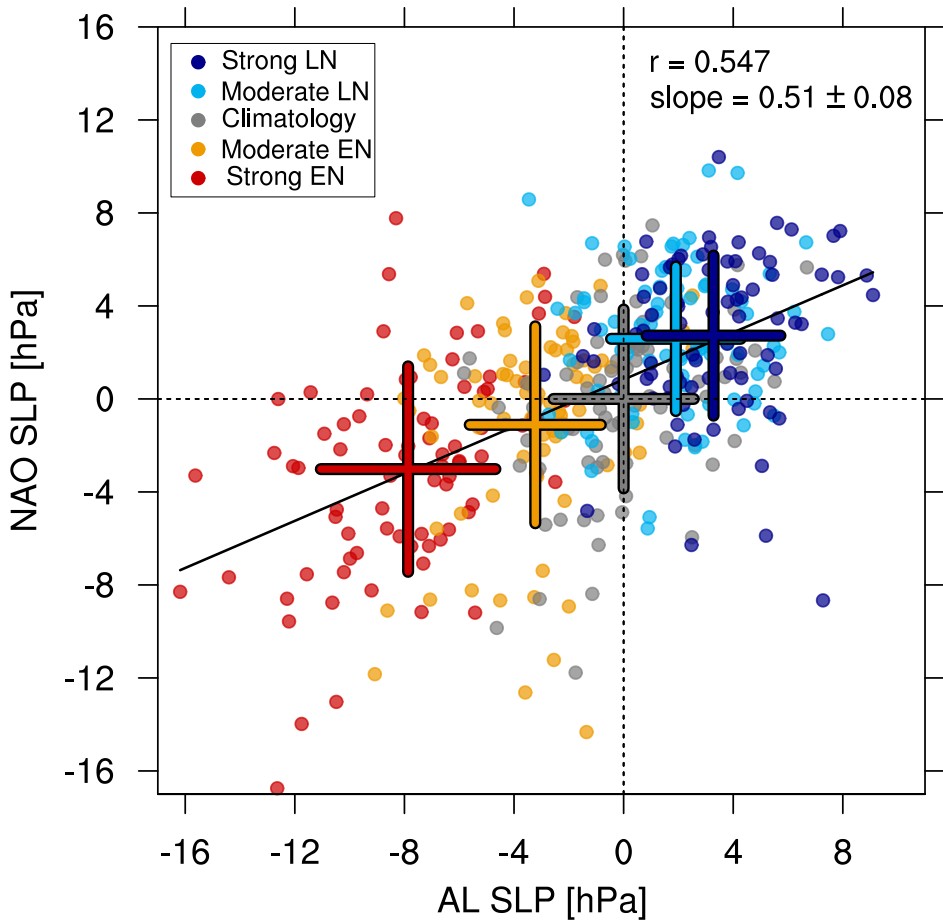

**Figure 8.** Scatter plot of the DJF mean NAO index versus the DJF mean AL index (not standardized) for all five experiments using nudging in the stratosphere (total of 396 years). Different colors identify the different ENSO forcings (red: strong EN, orange: moderate EN, grey: climatology, light blue: moderate LN, dark blue: strong LN). Crosses are centered at the mean values for each ENSO experiment, with the limits of the vertical and horizontal components corresponding to $\pm 1$ standard deviations of the NAO and AL indices respectively. The correlation coefficient (r) as well as the slope of the linear regression (black line) is shown at the top left corner. The slope error corresponds to a 95% confidence interval assuming a Gaussian distribution.

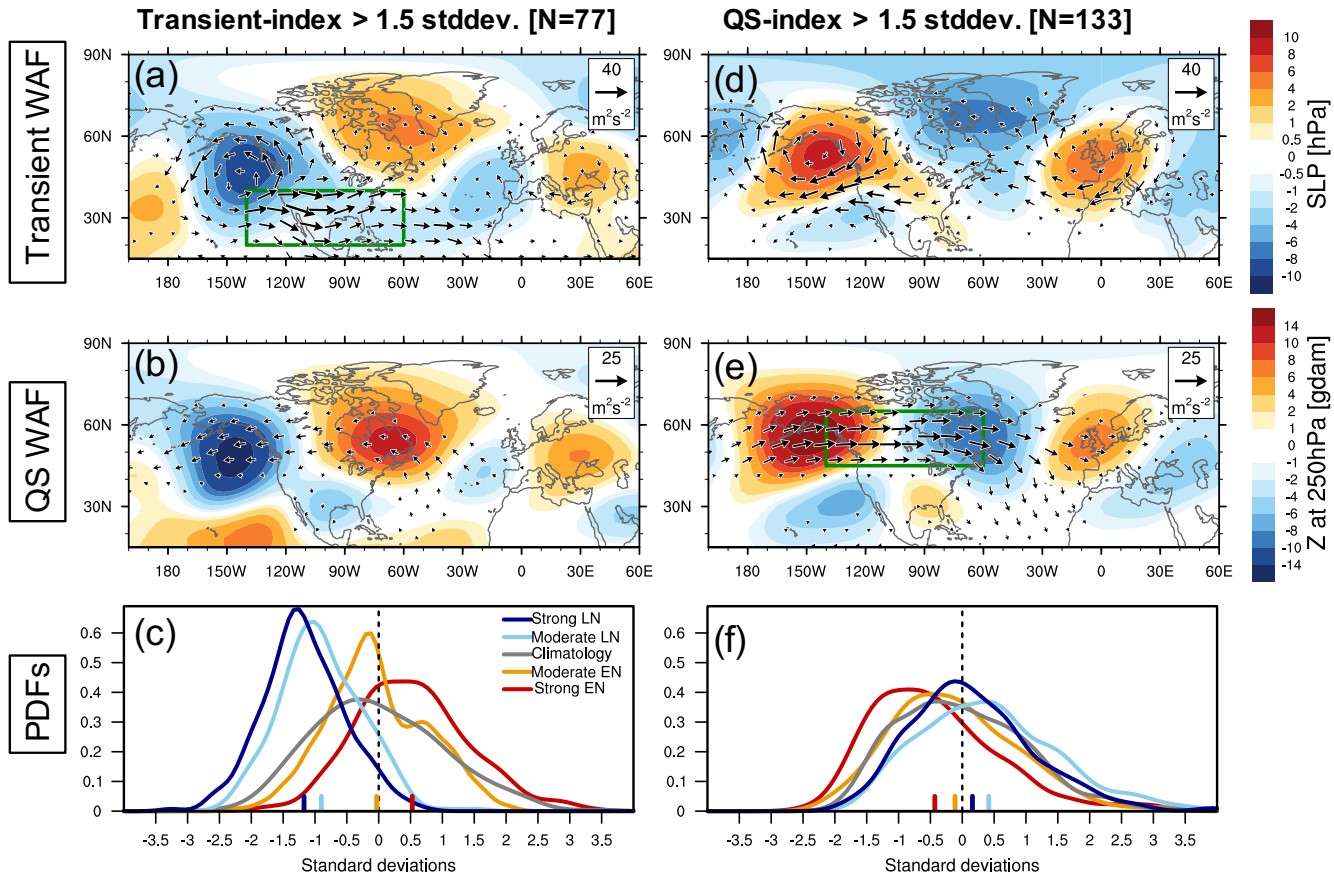

**Figure 9.** December to March monthly mean anomaly composites of (a,d) SLP (color shading) and transient WAF (arrows), and (b,e) Z250 (shading) and QS WAF (arrows), for months when anomalously strong eastward WAF of (a,b) transient eddies and (d,e) QS (k=1-3) waves occur. Events are defined using a threshold of 1.5 standard deviations for each of the indices. At the top, values in brackets indicate the total number of months considered in each composite. Green boxes indicate the regions where the eastward component of the WAF has been averaged for (a) transient and (e) QS waves. The PDF of the December to March standardized monthly means for (c) the $M_x$-index and (f) the $F_x$-index. Colors indicate different ENSO forcings (red: strong EN, orange: moderate EN, grey: climatology, light blue: moderate LN, dark blue: strong LN).

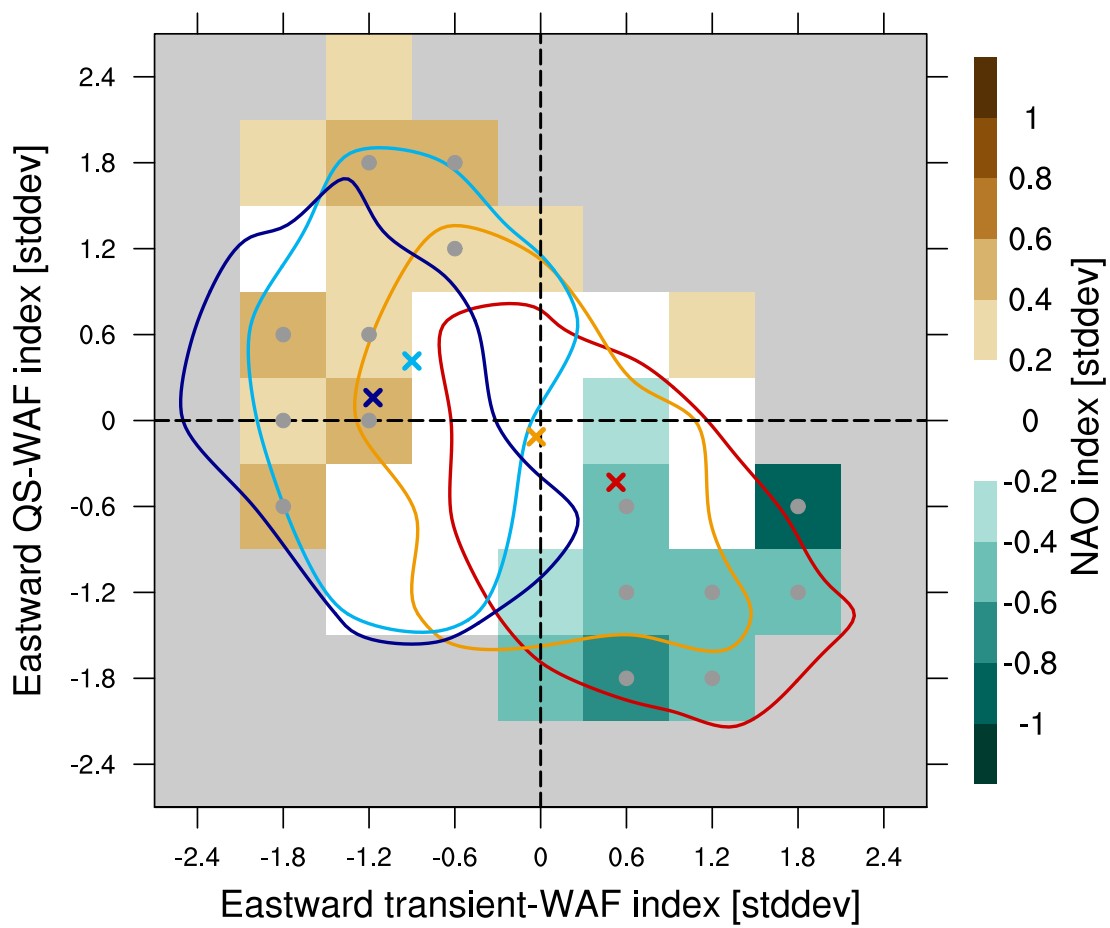

**Figure 10.** December to March monthly mean NAO index (color shading) as a function of the standardized eastward transient WAF ($M_x$) and the QS WAF ($F_x$) indices (definition in the main text) for the five simulations with stratospheric nudging (a total of 396 years). Grey dots in the middle of each cell indicate statistically significant values at the 95% confidence level according to a t-test with at least 10 data points. Light grey cells correspond to combinations of the WAF indices with less than 10 events. Colored contours represent the 2D-distribution of the number events for each ENSO experiment (red: strong EN, orange: moderate EN, light blue: moderate LN, dark blue: strong LN), where only the 10 events contour is shown for clarity, and where crosses indicate the mean values of the WAF indices for each of the ENSO forcing simulations.

**Table 1.** Frequency of extreme strong and weak eastward transient ($M_x$-index) and quasi-stationary ($F_x$-index) monthly WAF events. Asterisks * and double asterisks ** indicate that the mean value of the index distribution is statistically different from the climatological simulation at the 90% and 95% confidence level, respectively.

| Nudged stratosphere | Strong EN | Moderate EN | Climatology | Moderate LN | Strong LN |
|---|---|---|---|---|---|
| $M_x$-index $> 1.5$ stddev. | 13.3%** | 2.8% | 7.9% | 0.3%** | 0.3%** |
| $M_x$-index $< -1.5$ stddev. | 1.0%** | 2.2% | 5.0% | 16.8%** | 28.8%** |
| $F_x$-index $> 1.5$ stddev. | 5.1%** | 6.3%** | 6.3% | 15.5%** | 8.9%* |
| $F_x$-index $< -1.5$ stddev. | 12.3%** | 5.4%** | 3.8% | 0.3%** | 2.3%* |