# Peer review of "Nonlinearity in the Tropospheric Pathway of ENSO to the North Atlantic"

_Weather and Climate Dynamics, 2019_

## Referee Comment (RC1) · Anonymous Referee #1 · 5 Feb 2020

Review of " Nonlinearity in the Tropospheric Pathway of ENSO to the North Atlantic" by Bernat Jiménez-Esteve; Daniela I.V. Domeisen. Recommendation: minor revisions

The authors analyze the North Atlantic sector impacts of ENSO in simulations in which nudging is applied to the stratospheric circulation to shut down a stratospheric pathway. When the stratospheric pathway is shut down, the strongest –NAO response is achieved for the strong El Niño forcing, while the strongest +NAO like response is for a strong La Nina forcing. However the specific patterns are not linear, and the authors provide various diagnostics in order to explain the responses. The analysis is convincing, and nearly ready for publication.

Major comments: The authors note that the North Atlantic response to ENSO is much weaker than that in the Pacific sector, and they don't fully quantify this effect. Recent papers by Deser et al 2017, Garfinkel et al 2018, and Weinberger et al 2019 (all cited) perform a Monte Carlo analysis to compute how many events must be subsampled from all available model simulations before a given nonlinearity becomes statistically robust, and I think a similar analysis here would be helpful. If it turns out that e.g. >50 events are needed before nonlinearities become apparent, then such a nonlinearity may not be particularly useful for seasonal forecasting purposes. However this paper is publishable even if it turns out that nonlinearities are not large enough to be helpful.

Minor comments: 1. Line 79: That ENSO only accounts for some 10% of vortex variability was already pointed out by Garfinkel et al 2012. The correlation of Nino3.4 with seasonal mean vortex strength is $\sim$0.3 in reanalysis and also in a range of models (see their figure 5).

2. The studies conceptually most similar to the present one are Bell et al (2009) and Cagnazzo and Manzini (2009). While these papers are already cited, it would be helpful in the introduction to more explicitly discuss how the present analysis builds on this previous work.

3. Line 206 "Another interesting result is that "

4. Line 223: My understanding is that ISCA has a slab ocean at the bottom, not fixed SSTs. SSTs can be "specified" by running a tag-along Python script that computes the oceanic q-flux pattern that must be imposed in order to generate a desired SST pattern (Vallis et al 2018). If this is indeed the configuration the authors used, please clarify.

5. Line 405: "weak an asymmetric respect to" needs to be rewritten

6. Colorbar for figure 2: I find the units gdam confusing. Isn't this just dm?

7. The North Atlantic jet seems to be too zonally oriented in the climatology (see figure 3). This bias should be mentioned in the discussion section.

[Figure]

Garfinkel, C. I., Butler, A. H., Waugh, D. W., Hurwitz, M. M., and Polvani, L. M. ( 2012), Why might stratospheric sudden warmings occur with similar frequency in El Niño and La Niña winters?, J. Geophys. Res., 117, D19106, doi:10.1029/2012JD017777.

——————————————————

---

## Referee Comment (RC2) · Paloma Trascasa-Castro (Referee) · 5 Feb 2020

Summary:

The study addresses the tropospheric pathway of ENSO via the North Pacific to the North Atlantic using an idealised atmospheric model. They isolate the tropospheric pathway by relaxing stratospheric winds towards climatology and impose linearly increasing SST anomalies in the equatorial Pacific to simulate different magnitude El Niño and La Niña events. The study focuses on the role of quasi-stationary and transient waves for the propagation of the ENSO signal across North America and into the North Atlantic. While a nonlinear and asymmetric North Atlantic SLP response to

[Figure]

ENSO has been previously reported in the literature, the authors found that only their strong El Niño experiment produces a response that resembles a negative phase of the NAO, whereas similar SLP responses are observed for moderate and strong La Niña events, which are of comparable magnitude to the response to moderate El Niño events. The manuscript is clear and well written and reaches substantial conclusions that add knowledge to this area of study. The analysis of the model experiments is thorough and supports the main findings.

I do have some suggestions that I believe would clarify the interpretation of the results. The relationship between the Aleutian low and the North Pacific does not appear to entirely explain the North Atlantic response simulated in the model, and therefore my suggestion is to explore other routes of the tropospheric pathway of the ENSO-North Atlantic teleconnection such as the Caribbean Sea and the tropical North Atlantic. This would also help to put the results into the context of other recent studies focusing on the Caribbean Sea.

I consider the article suitable for publication in Weather and Climate Dynamics after clarifying and strengthening your argument on the comments below.

Recommendation: Minor revisions

General comments:

1. The study is explicit that it focuses on the North Pacific influence on the North Atlantic, but several studies highlight an important role for the tropospheric pathway via the tropical Atlantic (e.g., Toniazzo and Scaife, 2006; Hardiman et al., 2019; Ayarza-guÌLena et al., 2018). Though some discussion of this broader issue is given in the Conclusions, how important is the tropical Atlantic for the interpretation of the model results shown here? A tentative hint is given on line 265, but in my view the conclusions would be strengthened if this was made more explicit. Can you explain all of the North Atlantic/European response with the mechanisms put forth in section 6? If the model does not simulate a pathway via the Caribbean Sea is this a limitation of the

model? Or are there limitations of other studies that have argued for an important role for the tropical Atlantic pathway, e.g. they have neglected the North Pacific downstream effects?

2. Please be more consistent in the use of "linearity" and "asymmetry". I would suggest referring to "linearity" when you describe the dependence of the response on the magnitude of an ENSO event within the same phase (El Niño o La Niña), whereas when talking about asymmetry you compare the response to El Niño to the response to La Niña and assess whether the response to each ENSO phase is similar but opposite in sign. For example, Figures 5a) and b) shows asymmetry whereas Figures 5c) and d) shows nonlinearity.

Specific comments:

Lines 27-35 - For the non-expert reader it might help to include here a brief synopsis of what we know about the observed surface climate response to ENSO in Eurasia (temperature, precipitation).

Line 28 - Suggest adding reference (e.g. Li and Lau 2012) Li, Y., and N.-C. Lau, 2012: Impact of ENSO on the atmospheric variability over the North Atlantic in late winter—-Role of transient eddies. J. Climate, 25, 320–342, https://doi.org/10.1175/JCLI-D-11-00037.1

Line 30 - I think Bell et al. (2009) were earlier than these papers to distinguish the role of stratospheric and tropospheric pathways using experiments similar to those presented in this manuscript. I therefore suggest replacing these references or at least adding Bell et al. (2009).

Line 37 - Again, Bell et al. (2009) showed the influence of El Niño on SSWs before this paper.

Line 39 - Please keep the same methodology to determine the order of your citations, e.g. alphabetical, chronological or by degree of importance for supporting the previous

sentence.

Line 42 - You can reference here Table 2 of Trascasa-Castro et al. (2019) who provide a meta-analysis of studies of SSW changes under ENSO.

Line 43 - "longer time series" is vague - longer than what? The current reanalyses? What would constitute "long enough"?

Line 55 - Also Bell et al. (2009). Note also that Toniazzo and Scaife (2006) used a model that couldn't reproduce the stratospheric pathway of ENSO to the North Atlantic. A more recent reference that reaches a similar conclusion using a well resolved stratosphere model is Hardiman et al. (2019):

Hardiman, S. C., Dunstone, N. J., Scaife, A. A., Smith, D. M., Ineson, S., Lim, J., & Fereday, D. (2019). The impact of strong El Niño and La Niña events on the North Atlantic. Geophysical Research Letters, 46, 2874– 2883. https://doi.org/10.1029/2018GL081776

Line 60 - To distinguish from subtropical jet suggest: tropospheric = eddy-driven.

Lines 116-118 - I suggest strengthening the argument for why you impose opposite in sign but identical spatial pattern of SST anomalies. Garfinkel et al. (2018=the salience of nonlinearities...) suggested that the location of SST anomalies have a large influence, but as you said in Jimenez-Esteve and Domeisen (2019) the magnitude has a larger effect on the teleconnection than the spatial location of SSTs, and that's what you want to know.

Lines 134-136 - A bit more on how the nudging affects the tropospheric variability changes (or not) in the set-up used here would be helpful as compared to the control model. e.g., are there changes to the major modes of variability that go on to be assessed (e.g., amplitude of the NAO) and/or the tropospheric jet decorrelation timescale?

Line 183 - In agreement with Bell et al. (2009), Cagnazzo and Manzini (2009).

Line 186 - Is stronger than "is more than" and covers a larger area. That's really asymmetry rather than nonlinearity.

Line 186 - Refer to Figure 5 as well as to Figure 2d.

Line 192 - I don't see negative SLP anomalies in the North Atlantic as a response to moderate El Niño. I would rather say that moderate El Niño events only affect the Northern lobe of the NAO by leading to positive SLP anomalies of similar magnitude to strong El Niños. The SLP pattern shown in this work for strong El Niño (Fig. 2a) resembles the pattern shown by Toniazzo and Scaife (2006) in figure a20, correspondent to the El Niño event of 1998 which had a Niño3 SST anomaly of ∼2.7 K. Out of 20 events they examine, this is the only situation where positive SLP anomalies in the North Atlantic extend to Europe and negative SLP anomalies dominate in the southern lobe of the NAO (weakening the Azores high).

Lines 194-5 - A more detailed comparison of Figure 2a and 2b with Bell et al (2009) Figure 10 and Figure 11 middle right and lower right panels would also be helpful here as their experiments are for moderate and strong El Niño forcing with a degraded and relaxed stratosphere. There are some differences in your results, for example the location of the positive SLP anomaly in the North Atlantic in the moderate EN case; it would be instructive to the reader to discuss these more carefully as the comparison is very similar to your experiments.

Lines 208-220 - For a "pure" NAO- signal, one would expect the low-level Atlantic jet to shift south. For the experiment with the strongest projection onto NAO-, the strong EN case, the jet weakens rather than shifts (Fig 3a). For the weaker projections onto the NAO in the moderate EN and LN experiments the NA jet shows more of a shift. It therefore seems that the NAO does not fully explain the NA jet behaviour and low-level temperature patterns in the simulations. Have you thought about examining the East Atlantic pattern to see whether the response projects onto that mode (Figure 2b)?

It also seems (lines 216-219) you are saying to response is not barotropic, whereas a

pure internally generated NAO signal would typically shown an equivalent barotropic structure. Relevant to this point is the study by Mezzina et al (2020) so I suggest you include that as part of this discussion:

Mezzina, B., J. García-Serrano, I. Bladé, and F. Kucharski, 2020: Dynamics of the ENSO Teleconnection and NAO Variability in the North Atlantic–European Late Winter. J. Climate, 33, 907–923, https://doi.org/10.1175/JCLI-D-19-0192.1

Line 216 - Only a weak strengthening for La Niña.

Line 220 - Add references: Ineson and Scaife (2009), Cagnazzo and Manzini (2009).

Lines 223-225 - "For example, the weaker baroclinicity during strong EN tends to weaken the climatological Icelandic low, whereas the strengthening of the meridional temperature gradient during LN can be linked to the intensification of the Icelandic low and 245 the associated near surface westerly winds (Figure 3c,d)."

There is some nonlinearity here between baroclinicity anomalies over North America and the strength of the Icelandic low: For that specific low pressure system, and ignoring now the Azores high, baroclinicity anomalies are double in magnitude in strong ENSO events, whereas the strength of the Icelandic low seems the same in the moderate and strong events. Why is that?

Line 255 - Wave train pattern in figure 5a? Is a Rossby wave source anomaly plot necessary to identify possible sources in the Caribbean that might explain this NAO pattern?

Line 258 - There are other mechanisms through which ENSO can affect the NAO besides the one proposed in this study. In order to be able to explain the anomalous winds and temperature anomalies associated with both moderate and especially strong El Niño events, more analysis is necessary. I would suggest to plot Rossby waves source anomalies as well as SLP response by months to look for a non-stationary NAO response.

Garcia Serrano (2017) (https://doi.org/10.1175/JCLI-D-16-0641.1) studies the lagged ENSO-Tropical North Atlantic relationship which consist on a Gill-type response associated with a perturbed Walker Circulation. In your experiments SST are fixed so you cannot look at a lagged SST response in the TNA but you could look at the lagged SLP response in the North Atlantic, month by month as in Bell et al (2009) or Trascasa-Castro et al (2019) to see if there is any differences in the SLP response in the North Atlantic in late winter that might suggest an influence of the ENSO-TNA teleconnection as well as the ENSO-PNA teleconnection that you have described in your article.

Line 268 - Those studies suggest the dominance of the tropospheric pathway for strong EN is due to a saturation of the stratospheric pathway. However, in Hardiman et al (2019) their weak El Nino case shows a less active stratospheric pathway than observations which may highlight as issue with their approach. Trascasa-Castro et al (2019) showed the stratospheric pathway may not saturate for strong EN and hence there is still some debate around the proposed "saturation mechanism" which you should mention here.

Line 277 - I think this is a non-standard definition of the NAO index (neither station based nor EOF based). What are the implications of averaging over such a large area to calculate the NAO index rather than using Iceland and Azores?

Line 279 - Difficult to compare these (DJFM) with Fig 2 (DJF)

Lines 291-292 - I see what you are talking about, but is it partly a plotting issue? The amplitude of North Atlantic anomalies in Fig 6a is smaller than in 6b and you white out values <|0.5|hPa. If you add another contour does the dipole appear to extend further to Europe?

Lines 294-295 - Can you comment on whether there are changes to the shapes of the pdfs? It appears there might be so you could mention higher order moments than the mean if the differences are significant.

Line 293 - These other mechanisms could include the stratosphere but also the tropical Atlantic pathway; more analysis is needed to fully explain the positive SLP anomaly over Europe.

Line 303 - I don't agree that Fig 2b) shows that moderate EN projects onto a negative phase of the NAO. It might weakly project onto the NAO index as defined here, but it also looks like a blocking pattern.

Line 307 - Response might be lagged.

Line 314 - Is it remarkable? You did run the model for 80 years to get a high signal-to-noise ratio!

Line 316 - Trascasa-Castro et al. (2019) also show this result for the NAO so please add citation.

Lines 400-410 - While this synthesis of studies is useful some key points are missing:

⇢ Hardiman et al (2019) use ensemble of seasonal hindcasts, so the experiments are initialised and are individual ensemble members for only a few observed ENSO cases. This is a very different approach to the other atmospheric model studies described so is worth highlighting.

⇢ Rao and Ren (2016a) uses observations so is beset by small sample sizes, as you highlight as an issue on line 77

⇢ Weinberger et al (2019) use experiments with observed SSTs so their results capture differences in ENSO magnitude and pattern while this study, Rao and Ren (2016b) and Trascasa-Castro et al (2019) remove differences in pattern through an idealised experiment design.

Some editing of this paragraph to better clarify the above points would be helpful.

Lines 412-413 - Also likely to be important for determining how important the tropospheric and stratospheric pathways are would be the model's climatology in the stratosphere, e.g. Bell et al (2009) used a model with relatively few SSWs and Toniazzo and Scaife (2006) used a low top model with weak stratospheric variability.

Lines 432 - Add reference Rodriguez-Fonseca et al. (2016).

Lines 433 - Again mention this relies on a saturation of the stratospheric pathway for strong EN and it is still an open research issue as to whether this would occur. Even if the stratospheric pathway saturates at some point, its effect should no disappear altogether at strong EN as the results of Bell et al (2009) in their damped stratosphere case suggest.

Technical Corrections:

Line 17 - Define SST acronym in main text

Line 44 - "On average, Arctic stratospheric anomalies ..."

Line 59 - lead = leads

Line 74 - Therefore, the tropospheric pathway for ENSO impact

Line 152 - Previous = prior

Line 201 - Remove "do"

Line 314-315 - Remove "Figure 7 also serves to illustrate the large internal variability in the extratropics" = repeated sentence.

Line 321 - dominantly = predominantly

Line 401-402 - Replace "an state-of-the-art seasonal prediction model" with "atmospheric model" – the model is HadGEM3 which is in the same family as GloSea5 but run in a different configuration.

---

## Referee Comment (RC3) · Anonymous Referee #3 · 18 Feb 2020

This paper analyzes the tropospheric impacts of ENSO on the North Atlantic region, focusing on nonlinearities regarding the amplitudes of the events, and asymmetries comparing El Niño and La Niña phases. To do so, they use different idealized simulations with a simplified model in which stratospheric winds are nudged to climatological values to shut down the stratospheric ENSO pathway. General comments: This study extends that from Jimenez-Esteve and Domeisen 2019 who studied the nonlinearities of the ENSO teleconnections to the North Pacific. The authors use similar idealized experiments in both papers except that they shut down the stratospheric pathway by nudging the stratospheric winds to the climatology in the present study. For the reader's interest, it would have been easier to have one single paper on the asymmetries of the

tropospheric ENSO pathway and make the paper more self consistent and not having the reader go back and forth between papers. I detail below some of my concerns with comments to improve the paper, making it more self consistent and complete. Thus, I think the paper is appropriate for publication in Weather and Climate Dynamics after the authors address the comments below. I feel the discussion of mechanisms on the origins of the asymmetries, etc, is reduced to references to Jimenez-Esteve and Domeisen (2019). This is why in a few places I ask the authors to add more information to clarify certain aspects in addition to their reference to Jimenez-Esteve and Domeisen (2019). There are many references to Jimenez-Esteve and Domeisen (2019) but results from both papers are not really compared or discussed. Indeed, the comparison of the results could give us additional information not discussed in this study. The differences in the North Pacific between simulations in Jimenez-Esteve and Domeisen (2019) and the present manuscript must be related to the stratosphere (e.g. comparison of Figs. 2 and 5 in both papers). These differences and possible explanations should be discussed further not only in the Pacific (as it is done in Jimenez-Esteve and Domeisen (2019)) but also in the Atlantic. I also recommend plotting fig. 5 in the same projection as in Jimenez-Esteve and Domeisen (2019) for easy comparison. I see differences in the Atlantic region already comparing Fig.5 of both papers that need to be discussed. As figure 4 in Jimenez-Esteve and Domeisen (2019) compared their modeling results to reanalysis data, here a similar comparison should be made when possible, Figure 2 to 4 (or some of them) from Section 3 and figures 5 and 7 for asymmetries. Similarities and differences between model and reanalysis would give hints about how realistic are the modeling results when comparing the signals in the Pacific and about the relative role of the tropospheric and stratospheric pathways when comparing the signals in the Atlantic Ocean. Several studies have pointed out differences in the timing of the teleconnections in the Pacific and Atlantic Oceans (e.g. early vs late winter). How different are the responses and the non-linearities and asymmetries if we look at individual months or early vs late winter instead of DJF means? No need to show figures but add a sentence in the manuscript. Regarding wording, I find con-
fusing the use of 'nonlinearity' in certain places particularly in relation to 'asymmetry'. I recommend using nonlinearity for differences in the response regarding the magnitude of the events, and asymmetry for the differences between ENSO phases. Indeed, this is the way it was used in Fig. 5 in Jimenez-Esteve and Domeisen (2019) while here it is not. I find this confusing.

Minor comments: L. 30. I believe Bell et al. (2009), Cagnazzo and Manzini (2009) and Ineson and Scaife (2009) are the first ones to discuss the stratospheric pathway in connection to North Atlantic surface impacts. Please add the references. L.35-50. All this paragraph reads too long considering that the stratosphere is not the main focus of the study. I would shorten it and move it to line 75 to connect to the paragraph previous to the last one of this section. L. 53-L.70. The description of mechanisms is confusing. The authors start saying that they focus on the NP downstream effect. Which one is that of the ones described later on? Perhaps listing then as first, second, etc would help. L. 71-75. Which one is the mechanism used in the study? L.107. Please remove 'As in Jimenez-Esteve and Domeisen (2019)'.. it adds confusion. L.111. When mentioning here the four spatial patterns, please refer to figure 1a. L. 177. This first sentence is not very clear. Indeed the authors analyze these simulations throughout the paper. So the sentence can be improved to focus more on this particular section. Please substitute 'while relaxing. . .' by ' by relaxing . . .' L.180. Here and throughout the paper, are the results similar if we look at individual winter months instead of DJF averages? Do we see differences between early and late winter in the teleconnection and asymmetries? (see my general comments above). L.185 Following my comment about asymmetries vs linearity above, I think asymmetry should be used here. L. 232. Can the authors argue about why the response in temperature in EN over Europe is the opposite from a negative NAO? However, for LN the response is as expected, right? Section4 . Perhaps the title would be more appropriate as 'Spatial distribution of the asymmetry and non-linearity response to the ENSO. . .' or something similar. For a better comparison with Jimenez-ESteve and Domeisen (2019) please replot the figure with the same polar projection and add the same 'phase asymmetry' and 'single phase

nonlinearity' to the figure. L. 254. Where do we see in Fig. 5 that the asymmetry denotes a stronger AL/PNA for moderat EN than moderate LN? Individual phases are not shown here. L. 257 'EN compared to LN (not shown), and the strong . . .' L.266. Note also than in observations, the strongest EN winters are not accompanied by SSWs. L. 270. Note that the impact over Europe is linear (there is no signal in figure 5d). However, there is some positive signal in Fig. 5d in Jimenez-Esteve and Domeisen (2019), does this mean that the non linearity in that case came from the stratosphere? This is the type of comparison/discussion that needs to be included. L.298. and paragraph above. Can perhaps the authors explain a bit more about the origin of the nonlinearities (mechanism) here? L. 312. Can the authors include reanalysis data in Figure 7? Similar to the scatterplots in fig. 4 of Jimenez-Esteve and Domeisen (2019)? The comparison would give us a hint also on the role of the stratosphere. . . L. 19. Fig. 4 in Jimenez-Esteve and Domeisen (2019) do not show convection directly. Can the authors elaborate their argument a bit more here? L. 385. I also see a dipole for strong LN in Fig. 2d. L.388-390. I understand it might be difficult to answer, but the authors should discuss and elaborate on why moderate LN forcing has a stronger impact than moderate EN forcing or why there is a saturation effect for LN and not for EN? How all of this compares to observations? L. 392. Where is the sentence '. . .although the stratosphere may contribute when it is active' from? L. 400-410. I find this discussion too long for something not directly related to the paper, as there is no focus on the stratospheric nonlinearities. Please make it shorter.

―――――――――――――――――――――

---

## Author Comment (AC1) · 20 Apr 2020

The comment was uploaded in the form of a supplement:
http://www.weather-clim-dynam-discuss.net/wcd-2019-18/wcd-2019-18-AC1-supplement.pdf

---

## Author Response (AR1)

**Response to Reviewers**

We would like to thank the three reviewers for careful reading, insightful comments and helpful suggestions for our study. These have been included into the manuscript (see changes indicated in **bold** in the annotated manuscript attached at the end of the reviwer's response). Please find below the detailed responses (in blue) to the reviewers' comments and suggestions. All line indications refer to the new (annotated) version of the manuscript.

The main changes to the manuscript are listed here:

1. **Statistical Robustness of the nonlinearity and asymmetry**: A new figure has been added (figure 6), which quantifies the number of events that would have to be considered before the asymmetries and nonlinearities become statistically significant. This figure is based on a Monte Carlo approach similar to the one used by Garfinkel et al. (2018); Weinberger et al. (2019); Deser et al. (2017). This method is now also briefly described in the main text of the manuscript. Because of this, figures are re-numbered accordingly. Here we will refer to this new numbering.

2. **New supplementary material**: Following the reviewers' comments five new figures have been added to a new supplementary material. These figures include: The Rossby wave source response for the different experiments, the robustness of the SLP response as a function of the sample size (similar to Figure 5 but for each simulation), the first two EOF patterns in the North Atlantic sector in the JRA-55 reanalysis and in the model climatological simulations applying and not applying stratospheric nudging, a graphical representation of the 3 first moments of the AL and NAO PDFs shown in figure 7, and a scatter plot (similar to figure 8) showing the relationship between the NAO and AL using JRA-55 reanalysis.

3. **General text modifications**: Further details in response to the reviewers' comments have been added throughout the text. These include remarks on the effect of the stratospheric nudging, the sensitivity in SLP indices definitions and the intra-seasonal evolution of the response. This has lead to a substantial increase in the length of the manuscript. We have also made an effort to shorten the text where possible.

**Reviewer 1:**

Recommendation: minor revisions

The authors analyze the North Atlantic sector impacts of ENSO in simulations in which nudging is applied to the stratospheric circulation to shut down a stratospheric pathway. When the stratospheric pathway is shut down, the strongest –NAO response is achieved for the strong El Niño forcing, while the strongest +NAO like response is for a strong La Niña forcing. However, the specific patterns are not linear, and the authors provide various diagnostics in order to explain the responses. The analysis is convincing, and nearly ready for publication.

We thank the anonymous reviewer for their constructive comments, which have been addressed in the new version of the manuscript. We answer point by point to the comments below.

**Major comments:**

**1.** The authors note that the North Atlantic response to ENSO is much weaker than that in the Pacific sector, and they don't fully quantify this effect. Recent papers by Deser et al 2017, Garfinkel et al 2018, and Weinberger et al 2019 (all cited) perform a Monte Carlo analysis to compute how many events must be subsampled from all available model simulations before a given nonlinearity becomes statistically robust, and I think a similar analysis here would be helpful. If it turns out that e.g. >50 events are needed before nonlinearities become apparent, then such a nonlinearity may not be particularly useful for seasonal forecasting purposes. However, this paper is publishable even if it turns out that nonlinearities are not large enough to be helpful.

Thank you for this constructive comment. To address this concern we have added a new Figure in the manuscript (new Figure 6), following the visualization and method used in Garfinkel et al. (2018) and Weinberger et al. (2019). This figure shows the confidence intervals of the nonlinear and asymmetric SLP response when the winters in the simulation are sub-sampled in groups of increasing size. This analysis is done for four different area weighted SLP indices, i.e. the Aleutian low, the NAO, the Azores high and the Icelandic low. The definition of these indices is introduced in the main text (lines 346-348).

The bootstrapping methodology used here is the same as described in Garfinkel et al. (2018) and Weinberger et al. (2019) and consists in randomly selecting a sub-sample of a certain size from a pair of simulations to then computing the nonlinearity or asymmetric part of the response. This calculation is repeated 2000 times for different randomly selected sub-samples in order to obtain an estimation of the associated probability density function of the magnitude that we are interesting in determining (nonlinearity and asymmetry in our case). This technique is a generalization of the bootstrapping technique we already used to assess the significance of the response in Figures 1-4, with the only difference that the entire sample size (80 winters) of each simulation is randomly shuffled to find bootstrapped distributions of the response (ENSO

forcing - climatological simulations).

By successively increasing the size of the selected sub-samples we can answer the question of how many events must be considered before nonlinearities/asymmetries become statistically detectable at a certain confidence level (in our case 95%). This is done for the asymmetry and single phase nonlinearity both for strong and moderate ENSO events and it is shown in the new Figure 6, where the box plots display the 95 and 50% confidence intervals of the asymmetry and nonlinearity of the SLP response in the different predefined regions (shown in different colors). In the plot, when the whiskers (95% confidence interval) do not cross the zero-line the nonlinearity of that specific index becomes statistically detectable using the indicated number of events. The main results of this figure are now discussed in the text (lines 320-340).

Additionally, we have now employed the same Monte Carlo technique (but using the entire sample size, as in Figures 1 to 4) to display the significant asymmetry and nonlinearity at the 95% confidence level in Figure 5, which is consistent with the results from the new Figure 6.

**Minor comments:**

**1.** Line 79: That ENSO only accounts for some 10% of vortex variability was already pointed out by Garfinkel et al 2012. The correlation of Nino3.4 with seasonal mean vortex strength is 0.3 in reanalysis and also in a range of models (see their figure 5).
We agree with the reviewer about this point and therefore we have included this point and the reference in the text (lines 83-86).

**2.** The studies conceptually most similar to the present one are Bell et al (2009) and Cagnazzo and Manzini (2009). While these papers are already cited, it would be helpful in the introduction to more explicitly discuss how the present analysis builds on this previous work.
We now introduce the main findings of these two papers and how the analysis in the present paper builds on these results (lines 89-94). However, while the two studies mainly focus on the importance of the stratospheric pathway, the present study focuses on the tropospheric pathway and the nonlinearity of these responses in the NAE sector.

**3.** Line 206 "Another interesting result is that "
Changed

**4.** Line 223: My understanding is that ISCA has a slab ocean at the bottom, not fixed SSTs. SSTs can be "specified" by running a tag-along Python script that computes the oceanic q-flux pattern that must be imposed in order to generate a desired SST pattern (Vallis et al 2018). If this is indeed the configuration the authors used, please clarify.
In contrast to the previous studies by Thomson and Vallis (2018a,b), in the present study as well as in Jiménez-Esteve and Domeisen (2019) we do not use a slab ocean and therefore no q-fluxes, instead we prescribe observed climatological seasonally varying SSTs as stated in line 121 of the methods section. We have now included an extra remark to emphasize this (see line 117-118).

**5.** Line 405: "weak an asymmetric respect to" needs to be rewritten

Thanks for pointing this out. We have now corrected this sentence.

**6.** Colorbar for figure 2: I find the units gdam confusing. Isn't this just dm?

gdam = 'geopotential decameters $(10m)$'. In our understanding, dm would be decimeters $(0.1m)$. We added the gdam description in the caption of that figure for clarification.

**7.** The North Atlantic jet seems to be too zonally oriented in the climatology (see figure 3). This bias should be mentioned in the discussion section.

Yes, we agree with the reviewer about this model bias, which is common among many GCMs Zappa et al. (2013). We have now mentioned it in the results section as well as in the discussion (lines 499-500).

**Reviewer 2 (Paloma Trascasa-Castro):**

**Summary:**

The study addresses the tropospheric pathway of ENSO via the North Pacific to the North Atlantic using an idealised atmospheric model. They isolate the tropospheric pathway by relaxing stratospheric winds towards climatology and impose linearly increasing SST anomalies in the equatorial Pacific to simulate different magnitude El Niño and La Niña events. The study focuses on the role of quasi-stationary and transient waves for the propagation of the ENSO signal across North America and into the North Atlantic. While a nonlinear and asymmetric North Atlantic SLP response to ENSO has been previously reported in the literature, the authors found that only their strong El Niño experiment produces a response that resembles a negative phase of the NAO, whereas similar SLP responses are observed for moderate and strong La Niña events, which are of comparable magnitude to the response to moderate El Niño events. The manuscript is clear and well written and reaches substantial conclusions that add knowledge to this area of study. The analysis of the model experiments is thorough and supports the main findings.

I do have some suggestions that I believe would clarify the interpretation of the results. The relationship between the Aleutian low and the North Pacific does not appear to entirely explain the North Atlantic response simulated in the model, and therefore my suggestion is to explore other routes of the tropospheric pathway of the ENSO-North Atlantic teleconnection such as the Caribbean Sea and the tropical North Atlantic. This would also help to put the results into the context of other recent studies focusing on the Caribbean Sea.

I consider the article suitable for publication in Weather and Climate Dynamics after clarifying and strengthening your argument on the comments below.

**Recommendation: Minor revisions**

We thank Paloma Trascasa-Castro for her thorough and constructive comments on our manuscript, which have helped to improve this study. We answer point by point to her comments below.

**General comments:**

**1.** The study is explicit that it focuses on the North Pacific influence on the North Atlantic, but several studies highlight an important role for the tropospheric pathway via the tropical Atlantic (e.g., Toniazzo and Scaife, 2006; Hardiman et al., 2019; Ayarzaguena et al., 2018). Though some discussion of this broader issue is given in the Conclusions, how important is the tropical Atlantic for the interpretation of the model results shown here? A tentative hint is given on line 265, but in my view the conclusions would be strengthened if this was made more explicit. Can you explain all of the North Atlantic/European response with the mechanisms put forth in section 6? If the model does not simulate a pathway via the Caribbean Sea is this a limitation of the model? Or are there limitations of other studies that have argued for an important role for the tropical Atlantic pathway, e.g. they have neglected the North Pacific downstream effects?

We agree with the reviewer that the total North Atlantic response cannot be explained alone due to the downstream propagation of transient and quasi-stationary (QS) waves from the North Pacific as outlined in section 6 of the paper. However, the model response of the horizontal wave fluxes across North America explain reasonably well most of the nonlinearities observed in the North Atlantic response as shown in the old Figure 9 (new Figure 10). In the manuscript, we recognize that other pathways can contribute (see for example lines 54-56). In the present study we decided to focus on the North Pacific role, while leaving the contribution of the Caribbean and Tropical North Atlantic pathway for future work, while acknowledging its likely contribution.

Nonetheless, to answer the previous question, we have assessed if the model is able to simulate a pathway through the Caribbean Sea – Tropical Atlantic via a remote anomalous Rossby wave source in this region. We find that this can indeed be considered a relevant mechanism but mainly for the strong El Niño forcing (see the new supplementary Figure S1). Other studies (e.g., Ayarzagüena et al., 2018; Toniazzo and Scaife, 2006) have shown that the Tropical Atlantic pathway might be more relevant during early winter (ND) than late winter (JFM), when the tropospheric pathway through the North Pacific as well as the stratospheric pathway are fully developed. Therefore, here we mainly focus on the canonical winter response (DJF). We have now emphasized this in the introduction as well as in the discussion (lines 74-80 and 520-522).

**2.** Please be more consistent in the use of "linearity" and "asymmetry". I would suggest referring to "linearity" when you describe the dependence of the response on the magnitude of an ENSO event within the same phase (El Niño o La Niña), whereas when talking about

asymmetry you compare the response to El Niño to the response to La Niña and assess whether the response to each ENSO phase is similar but opposite in sign. For example, Figures 5a) and b) shows asymmetry whereas Figures 5c) and d) shows nonlinearity.

We thank the reviewer for this comment. Because in general an asymmetry can be considered a specific type of nonlinearity we sometimes use nonlinearity as a general term in the text, including both asymmetry and single phase linearity. However, we agree with the reviewer that this can sometimes be a bit confusing and thus we have now made this distinction whenever possible throughout the text (see mainly section 4).

**specific comments:**

**Lines 27-35** - For the non-expert reader it might help to include here a brief synopsis of what we know about the observed surface climate response to ENSO in Eurasia (temperature, precipitation).

Thanks, we have added a sentence in this regard (see lines 29-31).

**Line 28** - Suggest adding reference (e.g. Li and Lau 2012) Li, Y., and N.-C. Lau, 2012: Impact of ENSO on the atmospheric variability over the North Atlantic in late winter. The role of transient eddies. J. Climate, 25, 320–342, https://doi.org/10.1175/JCLI-D-11-00037.1

Thank you for the suggestion. We have now included this reference.

**Line 30** - I think Bell et al. (2009) were earlier than these papers to distinguish the role of stratospheric and tropospheric pathways using experiments similar to those presented in this manuscript. I therefore suggest replacing these references or at least adding Bell et al. (2009).

Thank you for noting this. We agree and therefore we have now included this reference.

**Line 37** - Again, Bell et al. (2009) showed the influence of El Niño on SSWs before this paper.

This reference has been included in this line.

**Line 39** - Please keep the same methodology to determine the order of your citations, e.g. alphabetical, chronological or by degree of importance for supporting the previous sentence.

We have now tried to be consistent and ordered the in-text citations by degree of importance and when they have equal importance we have ordered them chronologically.

**Line 42** - You can reference here Table 2 of Trascasa-Castro et al. (2019) who provide a meta-analysis of studies of SSW changes under ENSO.

We have now added a reference to Table 2 of Trascasa-Castro et al. (2019), which we agree that fits here very well.

**Line 43** - "longer time series" is vague - longer than what? The current reanalyses? What would constitute "long enough"?

We mean longer than the current reanalysis. We have now clarified this in the text.

**Line 55** - Also Bell et al. (2009). Note also that Toniazzo and Scaife (2006) used a model that couldn't reproduce the stratospheric pathway of ENSO to the North Atlantic. A more recent

reference that reaches a similar conclusion using a well resolved stratosphere model is Hardiman et al. (2019): Hardiman, S. C., Dunstone, N. J., Scaife, A. A., Smith, D. M., Ineson, S., Lim, J., Fereday, D. (2019). The impact of strong El Niño and La Niña events on the North Atlantic. Geophysical Research Letters, 46, 2874– 2883. https://doi.org/10.1029/2018GL081776

We have added these two references in the mentioned sentence.

**Line 60** - To distinguish from subtropical jet suggest: tropospheric = eddy-driven.

Thanks for the suggestion, we now refer to the tropospheric eddy driven jet.

**Lines 116-118** - I suggest strengthening the argument for why you impose opposite in sign but identical spatial pattern of SST anomalies. Garfinkel et al. (2018=the salience of nonlinearities...) suggested that the location of SST anomalies have a large influence, but as you said in Jimenez-Esteve and Domeisen (2019) the magnitude has a larger effect on the teleconnection than the spatial location of SSTs, and that's what you want to know.

Due to the design of the model experiments we isolate the nonlinearities and asymmetries originating only in the magnitude of the ENSO forcing, while removing the effect of the longitudinal location and asymmetry in the observed SST ENSO anomalies. However, we cannot say that the magnitude has a larger effect than the longitudinal location in causing such nonlinearities, but that these can originate solely due to the ENSO magnitude. In the real world, nonlinearities/asymmetries likely arise as a combination of these two factors (location and strength). Answering the question of which factor might be more important does not have a simple or a direct answer as these tend to be linked in the real world. In our study we focus on the nonlinearity/asymmetry of the response with respect the magnitude of the tropical forcing and therefore we cannot quantify if this is more or less important than the nonlinearities due to the location of the SST forcing. We have now clarified this in the methods section (lines 131-134).

**Lines 134-136** - A bit more on how the nudging affects the tropospheric variability changes (or not) in the set-up used here would be helpful as compared to the control model. e.g., are there changes to the major modes of variability that go on to be assessed (e.g., amplitude of the NAO) and/or the tropospheric jet decorrelation timescale?

Now we provide additional information about how the stratospheric nudging affects the tropospheric variability, which differs from our previous study Jiménez-Esteve and Domeisen (2019) which analyses the nonlinearity in the North Pacific sector but not using nudging.

In the nudging simulations employed in the present study, the interannual winter (DJF) NAO variance decreases by 40% with respect to the 5 identical simulations when the nudging is not applied. In the North Pacific, the effect of the nudging is much weaker than in the North Atlantic, and the Aleutian low variance decreases only by 10% when the nudging is applied. However, the applied stratospheric nudging does not lead to any significant circulation anomalies in the tropospheric mean flow and the main modes of variability (EOF based) in the North Atlantic remain unaltered. Therefore we concluded that the nudging technique is very useful to isolate the tropospheric pathway while it clearly removes the stratospheric pathway in the model simulations. This is shown in the new supplementary figure S3, which displays the first

two EOFs of the winter SLP in North Atlantic region for the JRA-55 reanalysis and for the model simulations with and without a nudged stratosphere. See changes in lines 152-159 in the annotated manuscript.

**Line 183** - In agreement with Bell et al. (2009), Cagnazzo and Manzini (2009).
We think these references are not needed here, the Aleutian low response to ENSO is a common feature reported in most of the cited studies in the introduction. These references are cited in several other places in our study.

**Line 186** - Is stronger than "is more than" and covers a larger area. That's really asymmetry rather than nonlinearity.
We have changed this sentence to "is stronger and covers a larger area than" and referred to it as an asymmetry instead of a nonlinearity.

**Line 186** - Refer to Figure 5 as well as to Figure 2d.
Thanks. We have included the reference to figure 5.

**Line 192** - I don't see negative SLP anomalies in the North Atlantic as a response to moderate El Niño. I would rather say that moderate El Niño events only affect the Northern lobe of the NAO by leading to positive SLP anomalies of similar magnitude to strong El Niños. The SLP pattern shown in this work for strong El Niño (Fig. 2a) resembles the pattern shown by Toniazzo and Scaife (2006) in figure a20, correspondent to the El Niño event of 1998 which had a Niño3 SST anomaly of 2.7 K. Out of 20 events they examine, this is the only situation where positive SLP anomalies in the North Atlantic extend to Europe and negative SLP anomalies dominate in the southern lobe of the NAO (weakening the Azores high).
We agree with the point of the reviewer and thus we have changed the following sentence in line 216: "the moderate EN forcing only leads to a significant response in the Northern lobe of the NAO, i.e. the Icelandic low, with an insignificant impact on the Azores high.". Furthermore we also added a reference to the Figure 2 in Toniazzo and Scaife (2006) and the similarity between the strong EN response and the 1998 EN event.

**Lines 194-5** - A more detailed comparison of Figure 2a and 2b with Bell et al (2009) Figure 10 and Figure 11 middle right and lower right panels would also be helpful here as their experiments are for moderate and strong El Niño forcing with a degraded and relaxed stratosphere. There are some differences in your results, for example the location of the positive SLP anomaly in the North Atlantic in the moderate EN case; it would be instructive to the reader to discuss these more carefully as the comparison is very similar to your experiments.
Thanks for pointing this out. We have added a few sentences in this paragraph to compare also the different response for moderate events in the two studies (lines 222-225). We think that this difference might be explained so to differences in the SST forcing. The forcing used in Bell et al. (2009) for EN is based on the average of the four strongest EN events. That leads to a forcing peaking above 2K in DJF which is stronger than the 1.5K imposed in our moderate EN simulations. This difference in the magnitude likely explains why their moderate EN response

pattern looks more like the strong EN response, just weaker in magnitude.

**Lines 208-220** - For a "pure" NAO- signal, one would expect the low-level Atlantic jet to shift south. For the experiment with the strongest projection onto NAO-, the strong EN case, the jet weakens rather than shifts (Fig 3a). For the weaker projections onto the NAO in the moderate EN and LN experiments the NA jet shows more of a shift. It therefore seems that the NAO does not fully explain the NA jet behaviour and low-level temperature patterns in the simulations. Have you thought about examining the East Atlantic pattern to see whether the response projects onto that mode (Figure 2b)?

We do not claim that the strong EN response "only" projects on the "pure" NAO-, and neither do the other three ENSO SLP responses. At lower levels the negative phase of the NAO can be in general both associated to a south shift and to a weakening of the zonal winds. See for example the Figure 11 in Woollings et al. (2010). This figure also shows that to fully describe the speed and latitudinal location of the North Atlantic low level eddy-driven jet, the NAO and the EA have to be used in combination. According to this same figure a weakening of the low level jet would correspond to a negative phase of the EA pattern. We now mention that the response for EN likely also projects onto the EA pattern (lines 249-250)

It also seems (**lines 216-219**) you are saying that the response is not barotropic, whereas a pure internally generated NAO signal would typically shown an equivalent barotropic structure. Relevant to this point is the study by Mezzina et al (2020) so I suggest you include that as part of this discussion: Mezzina, B., J. García-Serrano, I. Bladé, and F. Kucharski, 2020: Dynamics of the ENSO Teleconnection and NAO Variability in the North Atlantic–European Late Winter. J. Climate, 33, 907–923, https://doi.org/10.1175/JCLI-D-19-0192.1

We do not explicitly say that the response it not barotropic. In fact, we say it is mostly barotropic when comparing the geopotential and SLP response (see line 230). Nevertheless, we thank the reviewer for the suggested reference and we now cite Mezzina et al. (2020) in line 252 to say that a purely internally generated NAO would have a more barotropic structure.

**Line 216** - Only a weak strengthening for La Niña.

We have now emphasized that the response for La Niña is weak.

**Line 216 - Line 220** - Add references: Ineson and Scaife (2009), Cagnazzo and Manzini (2009).

In these lines we are explicitly referring to the result shown by Madonna et al. (2019) that the North Atlantic jet is more thermally driven during El Niño.

**Lines 223-225** - "For example, the weaker baroclinicity during strong EN tends to weaken the climatological Icelandic low, whereas the strengthening of the meridional temperature gradient during LN can be linked to the intensification of the Icelandic low and the associated near surface westerly winds (Figure 3c,d)." There is some nonlinearity here between baroclinicity anomalies over North America and the strength of the Icelandic low: For that specific low pressure system, and ignoring now the Azores high, baroclinicity anomalies are double in magnitude in strong ENSO events, whereas the strength of the Icelandic low seems the same in the moderate and

strong events. Why is that?

We agree with the reviewer that the relationship between baroclinicity and the Icelandic low pressure anomaly seems to be nonlinear. The relationship between this two magnitudes is rather complicated, it is therefore very difficult to infer any causality from just this two plots. Because there are different mechanisms involved in the tropospheric pathway, it can be that there is a compensation between the baroclinicity mechanism and the changes in the horizontal WAF for LN. In this regard, we have added some discussion in lines 315-319 saying that the nonlinearity within LN phase seems to be explained though the nonlinear response of the quasi-stationary eastward WAF (see the pdfs in the old Figure 8f, now figure 9f).

**Line 255** - Wave train pattern in figure 5a? Is a Rossby wave source anomaly plot necessary to identify possible sources in the Caribbean that might explain this NAO pattern?

As the reviewer points out, the asymmetry pattern in Figure 5a might be related to an asymmetry of the Rossby wave source (RWS) response in the Caribbean. In this regard, we have now included the Rossby wave source response for the 4 sensitivity experiments (Figure S1 in the supplementary). In Figure S1 an asymmetry between moderate EN and LN in the Caribbean-Tropical North Atlantic RWS is observed, which could be one of the factors together with the lack of a significant eastward transient WAF for the moderate EN simulation, explaining the wave-pattern in the North Atlantic SLP asymmetry for moderate events. This discussion has been now included in this section of paper (see lines 290-293)

**Line 258** - There are other mechanisms through which ENSO can affect the NAO besides the one proposed in this study. In order to be able to explain the anomalous winds and temperature anomalies associated with both moderate and especially strong El Niño events, more analysis is necessary. I would suggest to plot Rossby waves source anomalies as well as SLP response by months to look for a non-stationary NAO response.

We now included the mean DJF Rossby wave source response in the new supplementary Figure S1 and refer to it in the text (see answer to the reviewer comment above). The focus of the present paper is to characterize the canonical mid winter response (therefore we use DJF means for most of the analysis). Nonetheless, we have also checked the monthly evolution of the SLP response for each of the ENSO forced simulations, and we find that the response in the North Atlantic is strongest in December-January despite the strongest response in the North Pacific occurring in January-February. The weaker response in late winter might be explained due to the lack of the stratospheric pathway in the model experiments. We think the current simulation setup not adequate to provide further detail on the intra-seasonal evolution, as this we are missing important ingredients, like the remote SST anomalies in the tropical North Atlantic and North Pacific, and the stratospheric pathway. For this purpose a more complex model would be also desired and this is not the focus of the present study. This is highlighted in 284-285.

Garcia Serrano (2017) (https://doi.org/10.1175/JCLI-D-16-0641.1) studies the lagged ENSO-Tropical North Atlantic relationship which consist on a Gill-type response associated with a perturbed Walker Circulation. In your experiments SST are fixed so you cannot look at a

lagged SST response in the TNA but you could look at the lagged SLP response in the North Atlantic, month by month as in Bell et al (2009) or Trascasa-Castro et al (2019) to see if there is any differences in the SLP response in the North Atlantic in late winter that might suggest an influence of the ENSO-TNA teleconnection as well as the ENSO-PNA teleconnection that you have described in your article.

The lagged response of the Aleutian low SLP in the model is analysed in (Jiménez-Esteve and Domeisen, 2019). As mentioned above, the model experiments are too idealized to reproduce the lagged response in the North Atlantic. Differences between early and mid-to-late winter are not statistically different when considering the strong interannual variability in the North Atlantic circulation. Therefore to avoid confusion we decide to mainly focus on the winter averaged response.

**Line 268** - Those studies suggest the dominance of the tropospheric pathway for strong EN is due to a saturation of the stratospheric pathway. However, in Hardiman et al (2019) their weak El Nino case shows a less active stratospheric pathway than observations which may highlight as issue with their approach. Trascasa-Castro et al (2019) showed the stratospheric pathway may not saturate for strong EN and hence there is still some debate around the proposed "saturation mechanism" which you should mention here.

In the mentioned line we do not refer to the saturation of the stratospheric pathway because our model simulation exclude this mechanism by construction. According to our experiments the nonlinearity pattern observed in Figure 5a,c may also originate due to the competition between different nonlinear tropospheric 'pathways' or mechanisms, for example the Caribbean Rossby wave source and the downstream zonal wave propagation from the Pacific. We have now clarified this in lines 311-314.

**Line 277** - I think this is a non-standard definition of the NAO index (neither station based nor EOF based). What are the implications of averaging over such a large area to calculate the NAO index rather than using Iceland and Azores?

Averaging over a specific geographical area is a common approach in several modelling studies (e.g., Li and Wang, 2003; Stephenson et al., 2006; Zhang et al., 2019). Using an area average is better than using a point based index, as in model the centers of action might be slightly shifted with respect to observations. In addition, using EOFs to define the NAO variability leads to comparable results as is remarked in the text (line 348). We prefer to stick to the current definition as it is also more consistent to the Aleutian low definition and it allows us to divide the NAO into an Icelandic low and Azores high contribution. Figure S3 also shows the NAO pattern obtained using the EOF analysis and we can see that the pattern is identical to the one obtained using the averaged SLP difference between the Icelandic low and Azores high boxes (Figure 8).

**Line 279** - Difficult to compare these (DJFM) with Fig 2 (DJF)

We use monthly instead of seasonal anomalies as these better represent the sub-seasonal timescales on which these pressure systems vary, but using seasonal means leads to comparable results. This

allows us to also increase the sample size by a factor of 4, which enables us to obtain a better estimation of the respective PDFs. Find below the comparison between PDFs of the AL and the NAO using winter months and DJF means instead. Note that the results remain the same despite having a stronger signal in the DJF mean (color ticks along the x-axis).

[Figure]

Figure R1: The PDFs of the December-January-February-March standardized monthly means for (a) the AL and (c) the NAO indices. (b,d) the same but using DJF seasonal means instead. Colors represent the PDFs for the different ENSO forcings (red: strong EN, orange: moderate EN, grey: climatology, light blue: moderate LN, dark blue: strong LN).

**Lines 291-292** - I see what you are talking about, but is it partly a plotting issue? The amplitude of North Atlantic anomalies in Fig 6a is smaller than in 6b and you white out values $< |0.5|$ hPa. If you add another contour does the dipole appear to extend further to Europe? This is not a plotting issue, we have tried to add another contour and the dipole does not extend further much into Europe. The point we want to make here is that the influence of the AL over Europe is weaker because it is indirect, as compared to the more direct impact of the NAO.

**Lines 294-295** - Can you comment on whether there are changes to the shapes of the pdfs? It appears there might be so you could mention higher order moments than the mean if the differences are significant. We have now plotted a new figure S4, where the values of the 3 first moments (mean, standard deviation and skewness) are compared for the 5 simulations and for the AL and NAO monthly PDFs. A part from the already commented nonlinearity in the average response, we find that EN (LN) tends to increase (decrease) the standard deviation of both the AL and the NAO. A clear conclusion cannot be obtained for the skewness parameter, but we find that in general a strong EN decreases the negative skewness of the climatological NAO and makes the pdf more symmetric, similar to the AL. This Figure is now shown in the supplementary material, while in the manuscript we mention the previous observations. However, we do not have a clear

explanation for this higher momentum changes.

**Line 293** - These other mechanisms could include the stratosphere but also the tropical Atlantic pathway; more analysis is needed to fully explain the positive SLP anomaly over Europe.

We agree with the reviewer that other mechanisms have to be included to account for the full observed response over Europe. The tropical Atlantic contribution is not fully removed from this analysis, as the Aleutian low and the Rossby wave source in the Caribbean are themselves linked through the common ENSO forcing, while the stratospheric pathway is fully removed in our model experiments. We have now clarified this point in the text (see lines 311-314).

**Line 303** - I don't agree that Fig 2b) shows that moderate EN projects onto a negative phase of the NAO. It might weakly project onto the NAO index as defined here, but it also looks like a blocking pattern.

We agree with the reviewer that the moderate EN projects weakly on the negative NAO and that the response might project stronger on the East Atlantic pattern. However, in this section we address the linearity in the projection on the NAO pattern and thus introducing another index would complicate the interpretation of the results. Nevertheless, we have included a sentence on the text which says that the moderate EN in fact projects strongly on the second mode of variability in the North Atlantic sector.

**Line 307** - Response might be lagged.

This does not seem to be the case, as we use individual standardized monthly values. Moreover, the fact that the negative skewnees of the NAO is reproduced in the model is another prove that the model correctly reproduces the observed NAO variability.

**Line 314** - Is it remarkable? You did run the model for 80 years to get a high signal-to-noise ratio!

The variability of the NAO and the ENSO signal (thus the signal-to-noise ratio) do not vary with longer simulation times as this are intrinsic properties of the system. Due to the large internal variability of the NAO a large sample size is needed to obtain a significant correlation and that is what we did. In summary, while the long simulations make it possible to get a statistical robust value of the slope of the regression, that does not change the value of the slope. What we referred as to remarkable was the value of the slope, but we agree that the sentence is confusing and we have removed the statement.

**Line 316** - Trascasa-Castro et al. (2019) also show this result for the NAO so please add citation.

This citation has been added.

**Lines 400-410** - While this synthesis of studies is useful some key points are missing: - Hardiman et al (2019) use ensemble of seasonal hindcasts, so the experiments are initialised and are individual ensemble members for only a few observed ENSO cases. This is a very different approach to the other atmospheric model studies described so is worth highlighting.

- Rao and Ren (2016a) uses observations so is beset by small sample sizes, as you highlight as

an issue on line 77

- Weinberger et al (2019) use experiments with observed SSTs so their results capture differences in ENSO magnitude and pattern while this study, Rao and Ren (2016b) and Trascasa-Castro et al (2019) remove differences in pattern through an idealised experiment design.

We thank the reviewer for the above suggested points. We have now included this information points into the discussion section.

**Lines 412-413** - Also likely to be important for determining how important the tropospheric and stratospheric pathways are would be the model's climatology in the stratosphere, e.g. Bell et al (2009) used a model with relatively few SSWs and Toniazzo and Scaife (2006) used a low top model with weak stratospheric variability.

We also added the variability of the stratosphere as a factor to consider (line 499)

**Lines 432** - Add reference Rodriguez-Fonseca et al. (2016).

Thanks. We added this reference.

**Lines 433** - Again mention this relies on a saturation of the stratospheric pathway for strong EN and it is still an open research issue as to whether this would occur. Even if the stratospheric pathway saturates at some point, its effect should no disappear altogether at strong EN as the results of Bell et al (2009) in their damped stratosphere case suggest.

We now mention the saturation of the stratospheric pathway as a necessary condition for this result.

**Technical Corrections:**

Line 17 - Define SST acronym in main text changed

Line 44 - "On average, Arctic stratospheric anomalies ..." changed

Line 59 - lead = leads changed

Line 74 - Therefore, the tropospheric pathway for ENSO impact changed

Line 152 - Previous = prior changed

Line 201 - Remove "do" changed

Line 314-315 - Remove "Figure 7 also serves to illustrate the large internal variability in the extratropics" = repeated sentence. We modified the sentence

Line 321 - dominantly = predominantly changed

Line 401-402 - Replace "an state-of-the-art seasonal prediction model" with "atmospheric model" – the model is HadGEM3 which is in the same family as GloSea5 but run in a different configuration. changed

**Reviewer 3:**

This paper analyzes the tropospheric impacts of ENSO on the North Atlantic region, focusing on nonlinearities regarding the amplitudes of the events, and asymmetries comparing El Niño and La Niña phases. To do so, they use different idealized simulations with a simplified model in which stratospheric winds are nudged to climatological values to shut down the stratospheric ENSO pathway.

We thank the reviewer for their comments on our manuscript, which have helped to improve this study. We answer point by point to the comments below.

**General comments:**

**1.** This study extends that from Jimenez-Esteve and Domeisen (2019) who studied the non-linearities of the ENSO teleconnections to the North Pacific. The authors use similar idealized experiments in both papers except that they shut down the stratospheric pathway by nudging the stratospheric winds to the climatology in the present study. For the readers interest, it would have been easier to have one single paper on the asymmetries of the tropospheric ENSO pathway and make the paper more self consistent and not having the reader go back and forth between papers. I detail below some of my concerns with comments to improve the paper, making it more self consistent and complete.

Initially, as the reviewer suggests, we thought of publishing a paper with the results of the current paper and Jiménez-Esteve and Domeisen (2019) together. However, due to the quantity of interesting results and the different physical mechanisms in the North Pacific, we decided to split up the material into a paper focusing in the nonlinearity and asymmetry in the North Pacific and a more extensive paper focusing on the tropospheric pathway to the North Atlantic. Nonetheless, because the North Atlantic response to ENSO is influenced by the pathway through the North Pacific, we also have to refer to the response in the North Pacific in the current paper to explain the results here.

**2.** Thus, I think the paper is appropriate for publication in Weather and Climate Dynamics after the authors address the comments below. I feel the discussion of mechanisms on the origins of the asymmetries, etc, is reduced to references to Jimenez-Esteve and Domeisen (2019). This is why in a few places I ask the authors to add more information to clarify certain aspects in addition to their reference to Jimenez-Esteve and Domeisen (2019). There are many references to Jimenez-Esteve and Domeisen (2019) but results from both papers are not really compared or discussed. Indeed, the comparison of the results could give us additional information not discussed in this study.

We agree with the reviewer that we reference to our previous paper quite often in the current manuscript. This is of course mostly necessary as both studies use the same experiment design and similar methodology, and therefore it is easier and necessary to compare the results between these two studies. To facilitate the comparison between the two papers, we have added further

discussion comparing the model results, mainly in section 4, where the influence of the nudging of the stratosphere on the nonlinearities and asymmetries can be compared (see answer to the comments below).

**3.** The differences in the North Pacific between simulations in Jimenez-Esteve and Domeisen (2019) and the present manuscript must be related to the stratosphere (e.g. comparison of Figs. 2 and 5 in both papers). These differences and possible explanations should be discussed further not only in the Pacific (as it is done in Jimenez-Esteve and Domeisen (2019)) but also in the Atlantic. I also recommend plotting fig. 5 in the same projection as in Jimenez-Esteve and Domeisen (2019) for easy comparison. I see differences in the Atlantic region already comparing Fig.5 of both papers that need to be discussed.

To facilitate the comparison, we have now re-plotted figure 5 using a stereographic projection as in figure 5 in Jiménez-Esteve and Domeisen (2019). We can now see that most of the differences are localized in the North Atlantic, where the influence of stratospheric nudging is more relevant, while the nonlinearities in the North Pacific are similar for both plots. These differences are now mentioned in section 4

**4.** As figure 4 in Jimenez-Esteve and Domeisen (2019) compared their modeling results to reanalysis data, here a similar comparison should be made when possible, Figure 2 to 4 (or some of them) from Section 3 and figures 5 and 7 for asymmetries. Similarities and differences between model and reanalysis would give hints about how realistic are the modeling results when comparing the signals in the Pacific and about the relative role of the tropospheric and stratospheric pathways when comparing the signals in the Atlantic Ocean.

We agree with the reviewer that further comparison with reanalysis would ideally help. However, in Jiménez-Esteve and Domeisen (2019) it was already challenging to clearly compare the model results with observations, as very few strong ENSO events have been observed. This comparison becomes extremely challenging or impossible when analysing to the North Atlantic response in our model experiments, as the reviewer also points the 3 strongest EN events coincided with a stronger than normal stratospheric polar vortex (see for example Figure 3b,d in Hardiman et al. (2019)) and therefore we cannot conclude anything about the tropospheric pathway for strong events using simple composite techniques and reanalysis as these might also contain the anomalous stratospheric influence. Furthermore, as it is already discussed in the paper, there is still a big debate around the linearity in the stratospheric pathway in models and observations (Weinberger et al., 2019; Hardiman et al., 2019; Trascasa-Castro et al., 2019) and therefore we focus here on isolating the nonlinearity in the tropospheric pathway, which unfortunately can only be done using model simulations and not using reanalysis.

To show our point, below the JRA-55 reanalysis (Kobayashi et al., 2015) DJF SLP anomalies are plotted for the ENSO events of different magnitude. In this Figure shows that while in the North Pacific the response agrees quite well with model results, in the North Atlantic the response is different for the strongest events, and it projects onto the positive NAO phase, probably due to a destructive interference with the strong polar vortex observed during these events.

[Figure]

Figure R2: JRA-55 reanalysis (1958-2016) JFM mean SLP anomalies (in hPa) averaged over the (a) 4 strongest EN events (ONDJF Nino3.4 > 2 stddev.), (b) 8 moderate EN evnts ( 2 stddev. > ONDJF Nino3.4 > 1 stddev.), (c) 10 moderate LN events (-2 stddev. < ONDJF Nino3.4 < -1 stddev.), and (d) the 2 strongest Ln events (ONDJF Nino3.4 < -2 stddev.). Solid contours display the total averaged field.

In the case of figure 7 (now figure 8), we have tried to repeat the same analysis but using the JRA-55 reanalysis. This Figure now is shown in the supplementary material, as it is not really comparable to Figure 8. The figure shows a weaker and non-robust correlation between the DJF mean AL and the NAO indices. This might be due to the small sample size for strong ENSO events and the stratospheric influence, whose effect is removed on purpose in our model simulations. We refer to this result as well as to the explanation for the discrepancy in the main text (lines 391-395).

**5.** Several studies have pointed out differences in the timing of the teleconnections in the Pacific and Atlantic Oceans (e.g. early vs late winter). How different are the responses and the non-linearities and asymmetries if we look at individual months or early vs late winter instead of DJF means? No need to show figures but add a sentence in the manuscript.

We have analysed the winter monthly evolution of the response (November to March) and find that the response in the North Atlantic peaks around December-January depending on the simulation. In terms of its pattern the SLP and geopotential height response does not show significant intra-seasonal variations and remains similar to the averaged DJF response. The weaker response in late winter compared to reanalysis might be explained due to the lack of the stratospheric pathway in the model experiments. Therefore, we think that the current

simulation setup is not adequate to provide further detail on the intra-seasonal evolution, as this are missing important ingredients, like the remote SST anomalies in the tropical North Atlantic and the stratospheric pathway, the latter might be more important in late winter. For this purpose a more complex model would be desired and this is not the focus of the present study. This is now mentioned in the article (lines 284-285 and 521-522)

**6.** Regarding wording, I find confusing the use of 'nonlinearity' in certain places particularly in relation to 'asymmetry'. I recommend using nonlinearity for differences in the response regarding the magnitude of the events, and asymmetry for the differences between ENSO phases. Indeed, this is the way it was used in Fig. 5 in Jimenez-Esteve and Domeisen (2019) while here it is not. I find this confusing

We have now made figure 5 similar to figure 5 in Jiménez-Esteve and Domeisen (2019). See also response to reviewer 2: "Because in general an asymmetry can be considered a specific type of nonlinearity we sometimes use nonlinearity as a general term in the text, including both asymmetry and single phase linearity. However, we agree with the reviewer that this can sometimes be a bit confusing and thus we have now made this distinction whenever possible throughout the text"

**Minor comments:**

**L. 30**. I believe Bell et al. (2009), Cagnazzo and Manzini (2009) and Ineson and Scaife (2009) are the first ones to discuss the stratospheric pathway in connection to North Atlantic surface impacts. Please add the references.

Thanks for pointing this out. We have now included them in this line.

**L.35-50.** All this paragraph reads too long considering that the stratosphere is not the main focus of the study. I would shorten it and move it to line 75 to connect to the paragraph previous to the last one of this section.

Thank you for pointing this out, this paragraph has been shortened.

**L. 53-L.70.** The description of mechanisms is confusing. The authors start saying that they focus on the NP downstream effect. Which one is that of the ones described later on? Perhaps listing then as first, second, etc would help.

We agree with the reviewer that the beginning of the paragraph was a bit misleading. We have now listed and clarified the explanation of the tropospheric mechanisms.

**L. 71-75**. Which one is the mechanism used in the study?

We have now added the following sentence: "We focus on the effect of the changes in the total eastward wave activity fluxes of transient and QS waves, and the baroclinicity mechanism. However, nonlinearities in the North Atlantic response to ENSO are better explained in terms of the former mechanism."

**L.107**. Please remove 'As in Jimenez-Esteve and Domeisen (2019)'.. it adds confusion.

Removed

**L.111.** When mentioning here the four spatial patterns, please refer to figure 1a.

Thanks, we have added now this reference.

**L. 177.** This first sentence is not very clear. Indeed the authors analyze these simulations throughout the paper. So the sentence can be improved to focus more on this particular section. Please substitute 'while relaxing. . .' by ' by relaxing . . .'

Changed 'by relaxing'.

**L.180.** Here and throughout the paper, are the results similar if we look at individual winter months instead of DJF averages? Do we see differences between early and late winter in the teleconnection and asymmetries? (see my general comments above).

Yes, there are some small differences when looking at individual months. This is due to the fact that the peak of the response varies slightly between the different forcing experiments, December or January. However, the teleconnection patterns remain similar (see answer to the previous comments).To avoid confusion and unnecessary detail, in the manuscript we prefer to keep using DJF means to analyse the asymmetry and nonlinearity. This is stated now at the beginning of section 4.

**L.185** Following my comment about asymmetries vs linearity above, I think asymmetry should be used here.

Thanks, we now use here the term asymmetry.

**L. 232.** Can the authors argue about why the response in temperature in EN over Europe is the opposite from a negative NAO? However, for LN the response is as expected, right?

This is because, although EN projecting on a negative NAO, the pattern does not extend over the European continent as the full pattern of the NAO. Instead we find that positive geopotential anomalies occur over Europe during strong EN (Figure 2a). This leads to a warm air advection from the warmer North Atlantic ocean (Figure 3a). See lines 264-266.

**Section 4** . Perhaps the title would be more appropriate as 'Spatial distribution of the asymmetry and non-linearity response to the ENSO. . .' or something similar. For a better comparison with Jimenez-Esteve and Domeisen (2019) please replot the figure with the same polar projection and add the same 'phase asymmetry' and 'single phase nonlinearity' to the figure.

We have changed the title to: "Spatial pattern and statistical robustness of the nonlinear and asymmetric North Atlantic response to ENSO forcing". This section has substantially changed following the recommendation of reviewer 1, we now analyse the robustness of the observed patterns (see new figure 6). Also following the reviewer suggestion, we replotted Figure 5 accordingly.

**L. 254.** Where do we see in Fig. 5 that the asymmetry denotes a stronger AL/PNA for moderate EN than moderate LN? Individual phases are not shown here.

We do not show individual phases here, but the sum of the EN and LN response. The above conclusion is obtained by comparing figure 2b and 2c. Because the AL response to moderate EN is negative and the response to moderate LN is positive, the negative sign over the AL region

in the asymmetry plot denotes a stronger impact during moderate EN. We have clarified this in the text (line 290).

**L. 257** 'EN compared to LN (not shown), and the strong . . .'
It is shown in Figure 2a,d. We now refer to this figure in the text.

**L.266** Note also than in observations, the strongest EN winters are not accompanied by SSWs.
Yes, we are aware of this and now we also mention this in the text (line 313).

**L.270.** Note that the impact over Europe is linear (there is no signal in figure 5d). However, there is some positive signal in Fig. 5d in Jimenez-Esteve and Domeisen (2019), does this mean that the non linearity in that case came from the stratosphere? This is the type of comparison/discussion that needs to be included.
We do not see significant differences between figure 5d in this paper and 5d in Jimenez-Esteve and Domeisen (2019). The most evident differences are between Figures 5a,b between the two papers, i.e. the asymmetry. In this case we agree with the reviewer that the differences in the asymmetric response have to be linked to the differences in the stratospheric response. We mention this in lines 298-301 of the new version of the manuscript. However, because the employed model is not able to reproduce a very 'realistic' stratospheric pathway as in reanalysis a detailed interpretation between the two studies might be affected by the mentioned model 'bias'. Nonetheless we think than the model results in this study can be trusted when removing the stratospheric influence (figure 5).

**L.298. and paragraph above**. Can perhaps the authors explain a bit more about the origin of the nonlinearities (mechanism) here?
The convective response to the underlying SST is nonlinear mainly for large SST forcings in the tropical Pacific (see Figure 4 in Jiménez-Esteve and Domeisen (2019)). This is due to a threshold behavior for the occurrence of deep convection (e.g. Johnson and Kosaka, 2016). We have clarified this in the text while also adding the above reference.

**L. 312.** Can the authors include reanalysis data in Figure 7? Similar to the scatterplots in fig. 4 of Jimenez-Esteve and Domeisen (2019)? The comparison would give us a hint also on the role of the stratosphere.
This is a good suggestion. We have plotted a similar Figure 7, but using the JRA-55 reanalysis. These results are now shown as a supplementary Figure S5, because the two figures are not really comparable due to the lack of the stratospheric influence. The figure shows a weaker and non-robust correlation between the DJF mean AL and the NAO indices. This might be due to the small sample size for strong ENSO events and the stratospheric influence which is on purpose removed in our model simulations. We refer to this result as well as to the explanation for the discrepancy in the main text (lines 391-394)

**L. 319.** Fig.4 in Jimenez-Esteve and Domeisen (2019) do not show convection directly. Can the authors elaborate their argument a bit more here?
Figure 4 in Jiménez-Esteve and Domeisen (2019) shows the central Pacific poleward divergent

wind at 150hPa with respect to the NINO3.4 SST anomalies. This is a direct measure of the dynamical response within the tropics. Another reason to use divergent winds instead of OLR for example is because our model does not simulate clouds. Nevertheless, divergence at the upper troposphere in the tropics is very closely linked to the deep tropical convection. We have rewritten this sentence to clarify our reasoning.

**L. 385.** I also see a dipole for strong LN in Fig. 2d.
Yes, we agree with the reviewer that there is also a dipole structure, but the conclusion that the strongest impact is located over the Icelandic low does not change.

**L.388-390.** I understand it might be difficult to answer, but the authors should discuss and elaborate on why moderate LN forcing has a stronger impact than moderate EN forcing or why there is a saturation effect for LN and not for EN? How all of this compares to observations? This question is indeed very difficult to answer and cannot be fully resolved in this paper, neither using reanalysis where this asymmetry is so far not present nor detectable. Using the several diagnostics in this paper we can explain the stronger projection of the moderate LN response onto the NAO due to several factors:

- Stronger QS WAF response for moderate LN than for EN (see Figure 9f) and thus a stronger impact over the Icelandic low for moderate LN.

- The moderate EN leads to a stronger Rossby wave source response in the tropical North Atlantic (Figure S1), which might be related to the observed blocking pattern (Figure 2,b), which weakly projects onto the NAO.

- The insignificant response of the transient WAF for the moderate EN (Figure 9c) leads to an insignificant response over the Azores high, thus also weakly projecting onto the NAO.

Thus it seems that while the tropospheric mechanisms linking the AL availability and the NAO during the moderate LN are stronger, during moderate EN the tropical RWS in the Caribbean leads to a blocking pattern that weakly projects onto the NAO. These points are now included in the discussion section.

**L. 392.** Where is the sentence '. . .although the stratosphere may contribute when it is active' from?
Thanks, this part of the sentence has been removed.

**L. 400-410.** I find this discussion too long for something not directly related to the paper, as there is no focus on the stratospheric nonlinearities. Please make it shorter. We agree that the focus of the paper is the nonlinearities and asymmetries within the tropospheric pathway. However most of the previous literature on the North Atlantic response to ENSO consider the stratosphere as a crucial part to explain this teleconnection. Furthermore, we think that the disagreement between these studies is something that is worth mentioning in the discussion of the paper, which is one of the reasons that motivated us to exclude the stratospheric contribution

of this teleconnection. We have made an effort to shorten this paragraph, though as reviewer 2 recommends us to expand this paragraph by including several extra points, we have been working on finding a balance between these requests.

[revised manuscript text omitted]